# Aschoff's rule on circadian rhythms orchestrated by blue light sensor CRY2 and clock component PRR9

Yuqing He [1,2], Yingjun Yu [1,2], Xiling Wang [1,2], Yumei Qin [1,2], Chen Su [1,2] & Lei Wang [1,2] ✉

Circadian pace is modulated by light intensity, known as the Aschoff's rule, with largely unrevealed mechanisms. Here we report that photoreceptor CRY2 mediates blue light input to the circadian clock by directly interacting with clock core component PRR9 in blue light dependent manner. This physical interaction dually blocks the accessibility of PRR9 protein to its co-repressor TPL/TPRs and the resulting kinase PPKs. Notably, phosphorylation of PRR9 by PPKs is critical for its DNA binding and repressive activity, hence to ensure proper circadian speed. Given the labile nature of CRY2 in strong blue light, our findings provide a mechanistic explanation for Aschoff's rule in plants, i.e., blue light triggers CRY2 turnover in proportional to its intensity, which accordingly releasing PRR9 to fine tune circadian speed. Our findings not only reveal a network mediating light input into the circadian clock, but also unmask a mechanism by which the *Arabidopsis* circadian clock senses light intensity.

In order to adapt to the planet's rotation and orbital revolution around the sun, the evolutionarily conserved, self-sustaining circadian pacemakers must be continuously entrained by environmental signals to maintain the endogenous molecular and biochemical rhythms, and further adapt their rhythmic activities to the daily cycles[1–4]. In higher plants, the resonance of circadian pace with the daily changes in light signals increases competitive advantage and ensures optimal fitness in broad geographic locations, particularly across the latitudinal clines[5–8]. So, exploring the underlying mechanisms of how the plants sense periodic photoperiod changes season-by-season and changes in the light spectrum and intensity from day to night is paramount. Light is a predominant *zeitgeber* (time giver) signal that entrains the clock[9], and importantly light intensity is critical for circadian speed, well known as Aschoff's rule: high light intensity generally leads to lengthened circadian free-running periods in nocturnal organisms, while shortened periods in diurnal organisms including plants[10,11]. Surprisingly, how light intensity change is transmitted to the pacemaker is still a largely unresolved question in plants and animals.

Core oscillators of the *Arabidopsis* circadian clock include at least three interlocked transcriptional-translational feedback loops (TTFLs)[12–15]. The morning loop consists of reciprocally repressed members of dawn-expressed MYB domain transcription factors CIRCADIAN CLOCK ASSOCIATED 1 (CCA1) and LATE ELONGATED HYPOCOTYL (LHY), followed by morning-expressed PSEUDO-RESPONSE REGULATOR 9 (PRR9) and its family members PRR7[16,17]. Mechanistically, PRR9, peaking prior to PRR7 by about 2–3 h, recruits TOPLESS (TPL) and HISTONE DEACETYLASE 6/19 (HDA6/19) in vivo and forms a transcriptional co-repressor complex at *CCA1/LHY* promoter to trigger repression during the day[18]. PRR9 and PRR7 act redundantly not only as a crucial circadian output hub to control day-length dependent flowering time and growth dynamics at both transcriptional and post-translational levels[19–23], but also participate in the light input pathway[24–26]. Specifically, PRR9 mainly mediates blue light signal input while PRR7 acts in red light input[24]. Intriguingly, the dynamic phosphorylation level of PRR9, together with its protein abundance, reaches peaks at noon[25], which is positively correlated with natural light intensity fluctuation, implying PRR9 might serve a key role in

[1]Key laboratory of Plant Molecular Physiology, Institute of Botany, Chinese Academy of Sciences, Beijing 100093, China. [2]University of Chinese Academy of Sciences, Beijing 100049, China. ✉e-mail: wanglei@ibcas.ac.cn

transmitting blue light information to the core oscillator. However, the detailed mechanisms of PRR9 in mediating blue light input and post-translational modification of PRR9 are still unknown.

Environmental light information synchronizes and entrains the endogenous circadian clock, primarily through perception by photoreceptors. The flavin-containing photolyase-like photoreceptors CRYs, are unique as regulators of circadian that are evolutionarily conserved in both plants and metazoans[27,28]. In *Drosophila*, CRYs function as circadian photoreceptors by directly interacting with the core clock component TIMELESS (TIM) upon photoexcitation, resulting in the ubiquitination and subsequent rapid degradation of TIM protein[29], thus transmit blue light information to the circadian clock[30]. In mammalian and human, CRYs appear to be a transcriptional repressor of the clock itself[31–34]. In *Arabidopsis*, neither CRY1 nor CRY2 are core elements of the oscillator, instead, they were proposed to mediate blue light input redundantly[35–37]. Recently, CRYs were revealed to affect the circadian period through photo-signaling hub factors PIFs, COP1 or m6A methylation of over 10 circadian core oscillator genes (COGs)[4,38,39]. While it is still unclear whether CRY1 and CRY2 work in the same pathway to transmit a blue light signal to the core oscillator when considering distinct light-dependent stability of the respective proteins[40–42].

Here we report that photoexcited CRY2 but not CRY1 directly interacts with the core clock protein PRR9 in a blue light-dependent way. The physical interaction of CRY2 with PRR9 blocks the accessibility of the PRR9 protein to both its co-repressor TPL/TPRs and the kinase PPKs. Furthermore, the phosphorylation status of PRR9 is critical for its DNA binding ability and repression activity. Genetically, CRY2, but not CRY1, acts in the same pathway as PRR9 to regulate circadian clock modulation by blue light. Together, our findings elucidated the mechanism by which CRY2 and PRR9 interacts and gates blue light information into the circadian clock, which represents a key step for light input into the *Arabidopsis* circadian clock.

## Results

### PRR9 mediates blue light input to the clock by interacting with CRY2

To investigate the role of PRR9 in transmitting light information to the circadian clock, we initially analyzed the circadian phenotypes of *prr9-1* mutant and *CsVMVpro:PRR9-HA* over-expressing line (hereafter abbreviated as *PRR9ox-1*) in constant red (cR) and blue light (cB) conditions. By using *CCA1pro:LUC* as a circadian reporter, we found that under higher light irradiance (40 μmol m$^{-2}$ s$^{-1}$), both *prr9-1* mutant and *PRR9ox-1* displayed much more evident changes of circadian period in cB (Col-0 = 23.53 ± 0.09 h, *prr9-1* = 24.77 ± 0.1 h, *PRR9ox-1* = 22.99 ± 0.09 h) than in cR (Col-0 = 23.14 ± 0.05 h, *prr9-1* = 23.65 ± 0.04 h, *PRR9ox-1* = 23.03 ± 0.05 h) (Fig. 1a and Supplementary Fig. 1a), in line with the notion that PRR9 is more engaged in mediating blue light input to clock[24]. Furthermore, we examined the fluence response curve (FRC) of Col-0, *prr9-1* and *PRR9ox-1* in 2.5, 10, and 20 μmol m$^{-2}$ s$^{-1}$ condition and found *prr9-1* mutant showed more obvious elongated circadian period in cB than in cR in all tested light intensity (Fig. 1b). Interestingly, when light intensity increased from 2.5 to 10 μmol m$^{-2}$ s$^{-1}$, the slope of FRC for Col-0 (k = −0.32), *PRR9ox-1* (k = −0.48), and *prr9-1* mutant (k = −0.36) is in conformity with Aschoff's rule in cR, but *prr9-1* mutant against this rule in cB (k = 0.66) while the slope of FRC for Col-0 (k = −0.16) and *PRR9ox-1* (k = −0.18) in cB is similar with that in cR (Fig. 1b), further supporting the idea that the function of PRR9 is related with blue light intensity. Moreover, since either 1-h red or blue light pulse could dramatically induce the accumulation of *PRR9* transcripts, and neither *cry1* nor *cry2* mutant affected this acute change (Supplementary Fig. 1b), the specific role of PRR9 in the blue light input pathway could not be simply explained by its transcript level change. Hence, we were prompted to test if PRR9 could directly interact with CRYs to transduce blue light

information into the circadian clock. A direct and specific fluorescence signal was observed in the nuclei of cells co-expressing PRR9-nYFP with CRY2-cYFP but not with CRY1-cYFP or the negative control by bimolecular fluorescence complementation (BiFC) assay in the leaves of *Nicotiana benthamiana*. (Fig. 1c). The in planta interaction of PRR9 with CRY2 was further confirmed with split nano-luciferase complementation assay, in which bioluminescence signal was evidently detected when PRR9-NanoLucN (PRR9-NlucN) protein was transiently co-expressed with CRY2-NanoLucC (CRY2-NlucC) (Fig. 1d). Moreover, co-immunoprecipitation (Co-IP) assay showed that CRY2 but not CRY1 could specifically interact with PRR9 protein in vivo (Fig. 1e). To determine if PRR9/CRY2 interaction is light quality dependent, Co-IP assay was performed using infiltrated leaves of *Nicotiana benthamiana* exposed to dark, red, or blue light (10 μmol m$^{-2}$ s$^{-1}$) for 10 min, respectively. Remarkably, the treatment of blue light pulse appreciably enhanced the interaction between PRR9 and CRY2 compared to the dark treatment. In contrast, a red light appeared to reduce PRR9/CRY2 interaction (Fig. 1f, g). Consistently, Co-IP assay using etiolated seedlings of *CsVMVpro:PRR9-GFP* transgenic line also showed enhanced interaction between PRR9 and endogenous CRY2 under blue light condition (Supplementary Fig. 1c). Furthermore, in vitro pull-down assay demonstrated that PRR9-MBP beads had higher affinity with blue light-excited CRY2 than by red light-excited counterpart (Fig. 1h). Together, these results indicated that PRR9 specifically interacts with CRY2 but not CRY1 in a blue light-dependent manner, suggesting PRR9 mediates blue light input to circadian clock through physically associating with blue light receptor CRY2.

### CRY2 blocks the interaction of PRR9 with co-repressor TPL

To map the CRY2 domain required for interacting with PRR9, the GFP-tagged photolyase homology region (PHR) and C-terminal extension region (CCE) (Supplementary Fig. 2a) were respectively immunoprecipitated with PRR9-HA. Results showed that the PHR domain of CRY2 alone was sufficient to co-immunoprecipitate full-length PRR9 (Supplementary Fig. 2b). Reciprocally, PRR9 protein was divided into PRR9N, PRR9C, and PRR9D with a truncation of the repressive region (RR) containing EAR motif (Supplementary Fig. 2a), according to their distinctive functions[18,43–45]. Co-IP assay clearly demonstrated that CRY2 could interact with PRR9C but not PRR9N or PRR9D (Supplementary Fig. 2c, d), indicating the repressive region (RR) is required for PRR9 to interact with CRY2.

As the EAR motif of PRR9 within its repressive region is also necessary for interacting with transcriptional co-repressor TOPLESS (TPL), we asked if CRY2 would interfere with the interaction between PRR9 and TPL, thus compromising its transcriptional activity. In a competitive co-IP assay, we showed in the presence of CRY2, the interaction between PRR9 and TPL was greatly diminished compared to affinity ability in the absence of CRY2, as evidenced by a barely detectable co-immunoprecipitated TPL-FLAG signal (Fig. 2a), suggesting CRY2 competes with TPL for binding to PRR9. Consequently, the transcript levels of *CCA1* and *LHY*, two direct targets of PRR9, were reduced in *cry2* mutant, but were increased in *CRY2* overexpression transgenic lines *35S:GFP-CRY2* (*CRY2ox*) (Fig. 2b, c). As TPL can form complexes with HDA6/19 in vivo to facilitate the deacetylation process[46,47], we examined the levels of histone H3 lysine 9 acetylation (H3K9ac) on *CCA1* and *LHY* promoters by utilizing chromatin immunoprecipitation followed by quantitative PCR assay. We found that the H3K9ac levels in *CCA1* and *LHY* promoters in *cry2* mutant were slightly lower than those in Col-0 control plants (Fig. 2d, e). Moreover, the decreased level of H3K9ac in *cry2* could be reverted by *prr9-1*, as the H3K9ac levels on *CCA1* and *LHY* promoters in *cry2 prr9-1* double mutant were comparable to *prr9-1* single mutant (Fig. 2d, e), implying that PRR9 was required for CRY2-mediated change of chromatin status. We further examined the role of CRY2 on PRR9 transcriptional

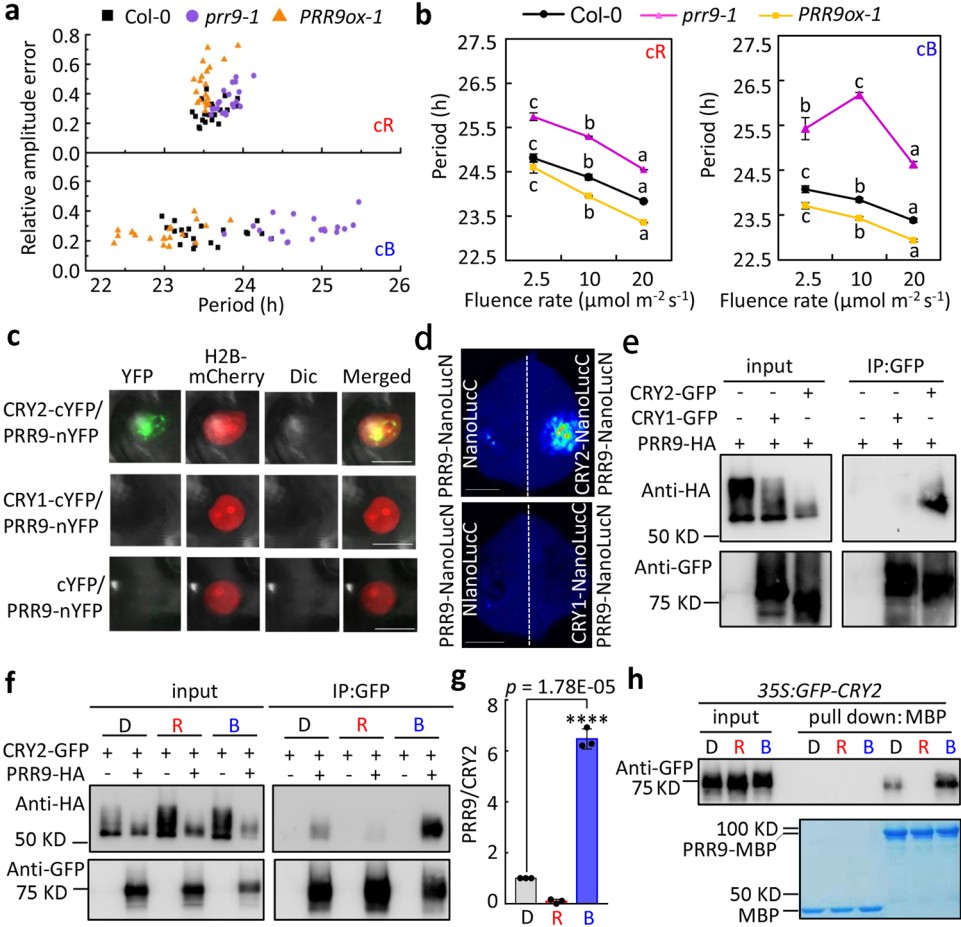

**Fig. 1 | CRY2 interacts with PRR9 in a blue light-dependent manner. a** Scatter plot showing circadian period and relative amplitude error of Col-0, *prr9-1*, and *CsVMVpro:PRR9-HA* (*PRR9ox-1*) seedlings in continuous red light and blue light (cB) (40 µmol m$^{-2}$ s$^{-1}$). Representative data from three biological repeats. **b** Estimated circadian period of *CCA1pro:LUC* in Col-0, *prr9-1*, and *PRR9ox-1* seedlings under cR (2.5 µmol m$^{-2}$ s$^{-1}$: $n = 60, 57, 56$; 10 µmol m$^{-2}$ s$^{-1}$: $n = 60, 57, 58$; 20 µmol m$^{-2}$ s$^{-1}$: $n = 60, 57, 58$ respectively) and cB (2.5 µmol m$^{-2}$ s$^{-1}$: $n = 61, 60, 60$; 10 µmol m$^{-2}$ s$^{-1}$: $n = 61, 60, 60$; 20 µmol m$^{-2}$ s$^{-1}$: $n = 62, 59, 60$, respectively). Lowercase letters indicate significant differences in the same genotypes among different light intensities through one-way ANOVA followed by Fisher's LSD test. Data represent mean ± s.e.m. from three biological repeats. **c** BiFC assay showing CRY2 but not CRY1 co-localizes with PRR9 in the nucleus. CRY1-cYFP, CRY2-cYFP, and cYFP were individually co-expressed with PRR9-nYFP in *N. benthamiana* leaves. H2B-mCherry as a nuclear marker. Bars, 20 µm. ($n = 10, 5, 10$ cells were examined/two leaves) **d** Nano-LUC-based split luciferase complemented assay showing the interaction of PRR9-NanoLucN with CRY2-NanoLucC but not with CRY1-NanoLucC. Representative data

from three independent experiments. Scale bar, 1 cm. **e** Co-IP assay showing the in vivo interaction of PRR9 with CRY2 but not CRY1 in *N. benthamiana* leaves. The IPed (CRY1 or CRY2) and co-IPed signals (PRR9) were detected by immunoblot with GFP and HA antibodies. All Co-IP assays were repeated at least three times. **f** Co-IP assay showing CRY2-GFP interacting with PRR9-HA in a blue light-dependent manner. After 3 days of transfection, the infiltrated *N. benthamiana* leaves were exposed to dark, red (R) and blue (B) light for 10 min, respectively. **g** Quantitative analysis of the densitometric ratio of co-IPed PRR9 to IPed CRY2 as shown in **f**. The interaction of PRR9 with CRY2 in dark conditions was set to 1 for normalization. Data were means ± s.d. from three biological replicates (****$p < 0.0001$ by one-way ANOVA). **h** In vitro pull-down assay of MBP or PRR9-MBP with GFP-CRY2 protein. Seven-day-old *35 S:GFP-CRY2* seedlings grown in LD (12 h light/12 h dark) condition were exposed to dark, R or B (10 µmol m$^{-2}$ s$^{-1}$) respectively for 10 min. GFP-CRY2 were detected using GFP antibody, MBP or PRR9-MBP were stained by coomassie brilliant blue R250 (CBB). Similar results were observed from three independent biological repeats.

repression activity in *N. benthamiana* system and *Arabidopsis* protoplast system by measuring the bioluminescence signals of *CCA1pro:LUC*. Evidently, the addition of CRY2 significantly compromised the transcriptional repression activity of PRR9 on the *CCA1* promoter, not that of the negative control *CsVMV* promoter (Fig. 2f, g). The quenching effect of CRY2 on PRR9 repressive activity on *CCA1* transcription was observed in *Arabidopsis* protoplast system as well (Fig. 2h). Together, the results indicated that CRY2 competes with TPL for binding to PRR9 protein, which reduces the transcriptional repression activity of PRR9 on its target genes such as *CCA1* and *LHY*.

**CRY2 inhibits PRR9 phosphorylation**

Noticeably, the ladder-like bands of PRR9-HA were diminished when co-expressing with CRY2 (Fig. 1e). Given PRR9 is a highly phosphorylated protein in *Arabidopsis*[25], we hypothesized that CRY2 could inhibit the

phosphorylation of PRR9. Results showed that the electrophoretic mobility shift of PRR9-HA or PRR9-GFP could be eliminated by lambda phosphatase (λPPase) treatment while blocked by phosphatase inhibitors NaF/Na$_3$VO$_4$ (Fig. 3a, b), suggesting the band shift of PRR9 was indeed caused by phosphorylation. Since there was only a single PRR9 band in the presence of CRY2 (Fig. 3b), the results suggest that CRY2 could efficiently inhibit PRR9 phosphorylation. As CRY2 protein is relatively stable in low blue light but is subjected to rapid degradation in high irradiance of blue light[40–42], we then examined the phosphorylation pattern of PRR9 under a range of blue light intensity by using a previously characterized *PRR9:PRR9-GFP* transgenic line[25]. Seedlings of *PRR9:PRR9-GFP* were exposed to high, medium and low (80, 20, and 1 µmol m$^{-2}$ s$^{-1}$) blue light at *zeitgeber* time 0 (ZT0). The protein abundance of CRY2 was rapidly reduced in the high irradiance of blue light. By contrast, the PRR9 protein was appreciably accumulated in high blue

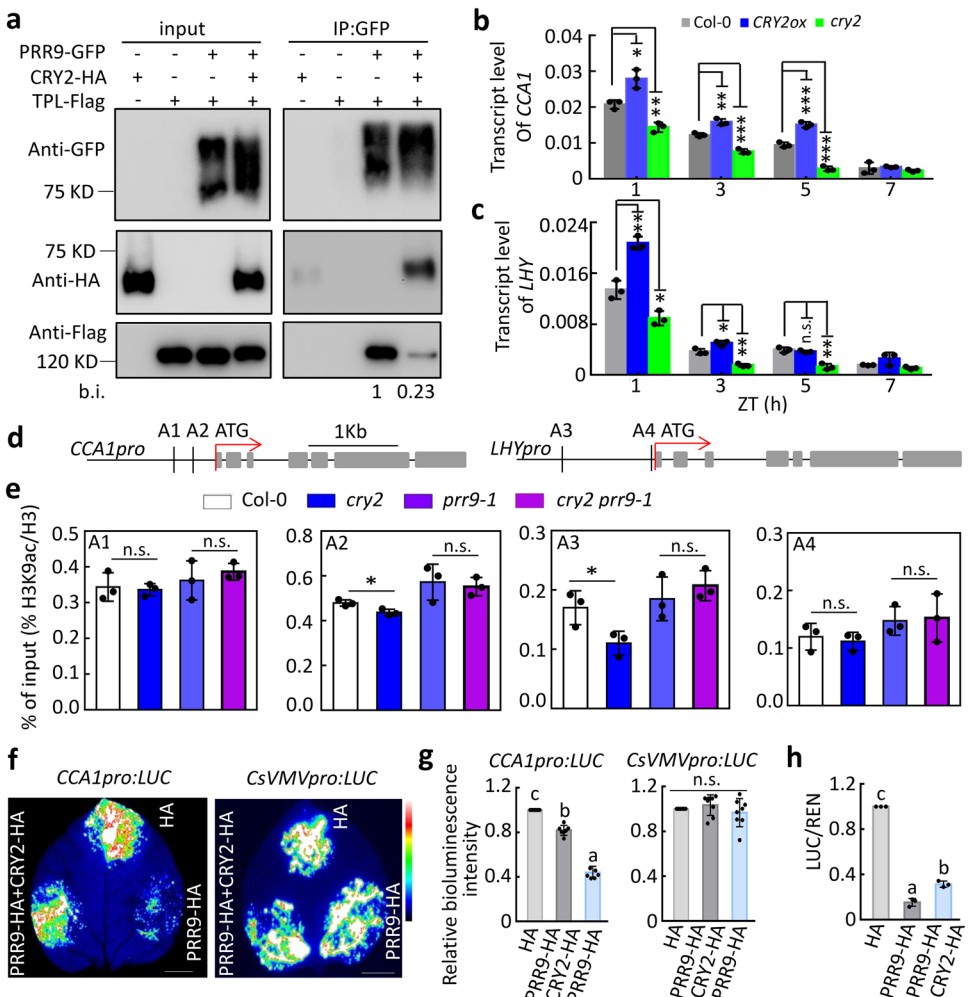

**Fig. 2 | CRY2 inhibits the transcriptional repressive activity of PRR9. a** Co-IP assay showing CRY2 interfered with the interaction of PRR9 with TPL. The IPed (PRR9-GFP) and co-IPed signals (TPL and CRY2) were detected by immunoblot with GFP, Flag and HA antibodies, respectively. b.i. stands for band intensity. **b**, **c** Temporal expression level of *CCA1* (**b**) and *LHY* (**c**) in Col-0, *35S:GFP-CRY2* (*CRY2ox*), and *cry2* mutant by RT-qPCR. Seedlings were grown for 10 days in LD condition and then exposed to 2.5 μmol m$^{-2}$ s$^{-1}$ blue light for indicated time points. Data were mean ± s.d., $n = 3$, technical repeats. ***$p < 0.001$, **$p < 0.01$, *$p < 0.05$, n.s. means no significant difference compared to Col-0 by two-tailed student's *t*-test. Similar results were observed from three independent repeats. **d** Simplified schematic representing promoter regions and gene structures of *CCA1* and *LHY*. A1 and A2 denote amplicon regions −516 to −303 and −313 to −90 on *CCA1* promoter, A3 and A4 denote amplicon regions −1075 to −931 and −44 to +192 on *LHY* promoter. **e** Relative H3K9ac/H3 levels in Col-0, *cry2-1*, *prr9-1*, and *cry2-1 prr9-1*. Two weeks of seedlings entrained in LD cycles were harvested at ZT4 for ChIP assay with H3K9ac or H3 antibody. The enrichment of amplicons was analyzed by ChIP-qPCR. Data

were mean ± s.d., $n = 3$, technical repeats. Similar results were observed from two biological repeats. *$p < 0.05$, n.s. means no significance between groups by two-tailed student's *t*-test. **f** Transient transcriptional expression analysis in *N. benthamiana* leaves shows that the repression activity of PRR9 is diminished by CRY2. *CCA1pro:LUC* and *CsVMVpro:LUC* were used as reporters. GFP and PRR9-GFP alone or with CRY2 were used as effectors, respectively. Scale bar, 1 cm. **g** Quantitative analysis of the *CCA1pro:LUC* and *CsVMVpro:LUC* bioluminescence is shown in **f**. Data were mean ± s.d., $n = 7$ and 8, respectively. Lowercase letters indicate significant differences while n.s. means no significant difference was determined by one-way ANOVA followed by Fisher's LSD test. **h** Quantification of relative LUC/REN activity showing CRY2 inhibits PRR9 repressive activity on *CCA1pro:LUC* in *Arabidopsis* protoplast. Data were mean ± s.d. $n = 3$, independent biological repeats by one-way ANOVA followed by Fisher's LSD test. Lowercase letters indicate significant differences. The value of relative LUC/REN activity in the protoplast transformed with control plasmids was defined as 1.

light (Fig. 3c). Moreover, the band shift of PRR9-GFP could be clearly detected in the medium and high blue light, especially at ZT5 and ZT7, but not in the low blue light (Fig. 3c). Besides, a significant increase of *PRR9* transcript was caused by high but not low blue light treatment (Fig. 3d). Taken together, both transcription and post-translational modification of PRR9 were modulated by blue light intensity.

To further examine the biological significance of CRY2-mediated inhibition of PRR9 phosphorylation, the in vivo phosphorylation sites of PRR9 protein were mapped with immunoprecipitation-liquid chromatography-mass spectrometry (IP-MS) analysis by utilizing *PRR9:PRR9-GFP* transgenic seedlings harvested at ZT5 when PRR9 abundance and phosphorylation level reach peak (Fig. 3e). Total

coverage of PRR9 protein was about 64.39% in three biological replicates (Supplementary Fig. 3a), and four phosphorylated peptides were identified (Fig. 3f and Supplementary Fig. 3b–d). Collectively, nine putative phospho-residues of PRR9, including Ser (S) 267, S269, S309, Thr (T) 310, T334, T335, S336, S337, and S365 were identified (Fig. 3g). Among the identified phosphorylation sites, S267 and S269 located within RR domain close to the known EAR motif (Fig. 3g). Subsequently, all detected phosphorylation sites were mutated to either non-phosphomimetic alanine (A) (PRR9$^{9A}$) or phosphomimetic aspartic acid (D) (PRR9$^{9D}$) tagged by GFP. As expected, PRR9 migrates faster than PRR9$^{9D}$ and slower than PRR9$^{9A}$, in which the upper phosphorylated band signal was hardly detected (Fig. 3h), indicating that

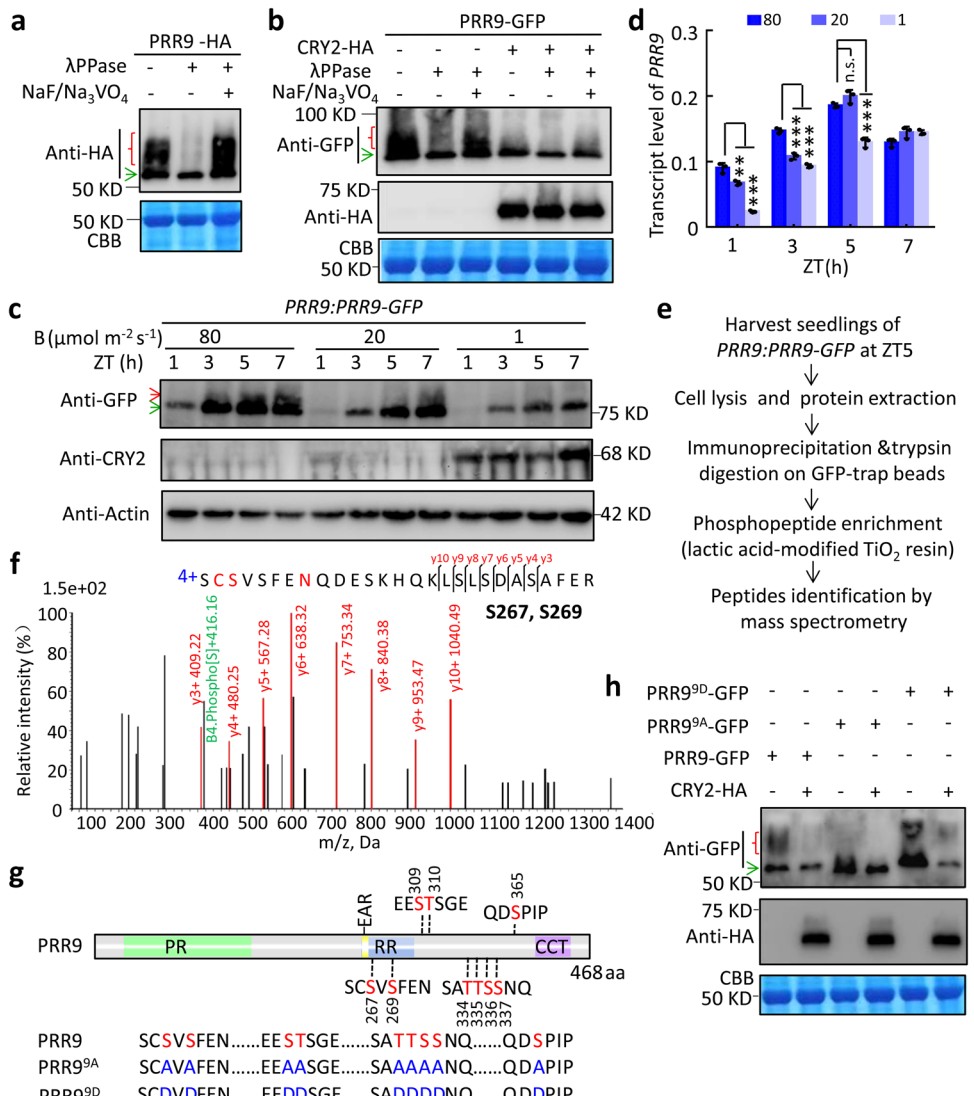

**Fig. 3 | CRY2 inhibits phosphorylation of PRR9 in a light intensity-dependent manner. a** Immunoblot showing the band shift of PRR9-HA was eliminated upon λPPase treatment. Total PRR9-HA tobacco protein lysates were treated with λPPase or phosphatase inhibitors NaF and Na₃VO₄, as noted. Red open brace and green arrow indicated phosphorylated and unphosphorylated forms of PRR9, respectively. CBB staining was used as a loading control. **b** Band shift of PRR9-GFP was largely reduced when co-expressing with CRY2-HA. Total tobacco protein lysates were treated as in **a**. **c** Immunoblot showing dynamic phosphorylation of PRR9 was regulated by blue light intensity. Ten days *PRR9:PRR9-GFP* transgenic plants were exposed to 80, 20 or 1 µmol m⁻² s⁻¹ blue light and harvested at indicated time points. Lysates were analyzed by immunoblot with GFP, CRY2, and Actin antibody. Red and green arrows indicated phosphorylated and unphosphorylated form of PRR9, respectively. **d** Transcriptional expression level of *PRR9* of seedlings in **c** by RT-qPCR. Data were mean ± s.d., *n* = 3, technical repeats. **$p < 0.01$, ***$p < 0.001$, n.s.

means no significant difference compared to 80 µmol m⁻² s⁻¹ by two-tailed student's *t*-test. **e** Schematic overview of PRR9 immunoprecipitation-liquid chromatography-mass spectrometry (IP-MS) workflow. Two-week-old *PRR9:PRR9-GFP* seedlings grown in LD condition were exposed to 40 µmol m⁻² s⁻¹ blue light for 5 h before harvesting. ***$p < 0.001$, **$p < 0.01$, *$p < 0.05$ by two-tailed student's *t*-test. **f** Representative mass spectrogram of PRR9 phosphopeptide. Phosphosites-S267/S269 were indicated. **g** Schematic diagram showing domain structure of PRR9 and position of phosphosites identified by IP-MS. The phospho-serine (S) or threonine (T) residues are highlighted with red letters, and mutation to alanine (A) or aspartic acid (D) were indicated with blue letters. **h** PRR9-GFP, PRR9^9A-GFP and PRR9^9D-GFP were transiently expressed with or without CRY2-HA in *N. benthamiana* leaves. Total lysates were analyzed by immunoblot probed with GFP and HA antibodies, respectively. Representative figures of **a**–**d** and **g** from three biological repeats.

the identified residues are major phosphorylation sites of PRR9. Moreover, the upper band of PRR9 and PRR9^9D was largely reduced in the presence of CRY2 (Fig. 3h), further verifying that CRY2 inhibits PRR9 phosphorylation.

## Phosphorylation of PRR9 is critical for its clock function

Phosphorylation is a critical and evolutionarily conserved post-translational modification for fine-tuning the circadian clock in many eukaryotes, which can affect protein stability, subcellular localization, and transcriptional activity[48–50]. To evaluate the effects of PRR9 phosphorylation status on circadian clock function, we generated

transgenic lines stably expressing S267A/S269 A, S309A/T310A, T334A/T335A/S336A/S337A, S365A, 9A and 9D, driven by PRR9 native promoter in the *prr9-1* background. The expression of *PRR9* was substantiated in two independent transgenic lines by protein immune-blot and RT-qPCR assay, respectively (Supplementary Fig. 4a–d). Firstly, we found that the higher transcript levels of *CCA1* and *LHY* in the *prr9-1* mutant could be fully complemented by phosphomimetic PRR9^9D but not non-phosphomimetic PRR9^9A. In addition, S309A/T310A, T334A/T335A/S336A/S337A, and S365A could also effectively decrease the transcript levels of *CCA1* and *LHY* in the *prr9-1* mutant. By contrast, phospho-mutation of S267A/S269A in the RR motif failed to fully

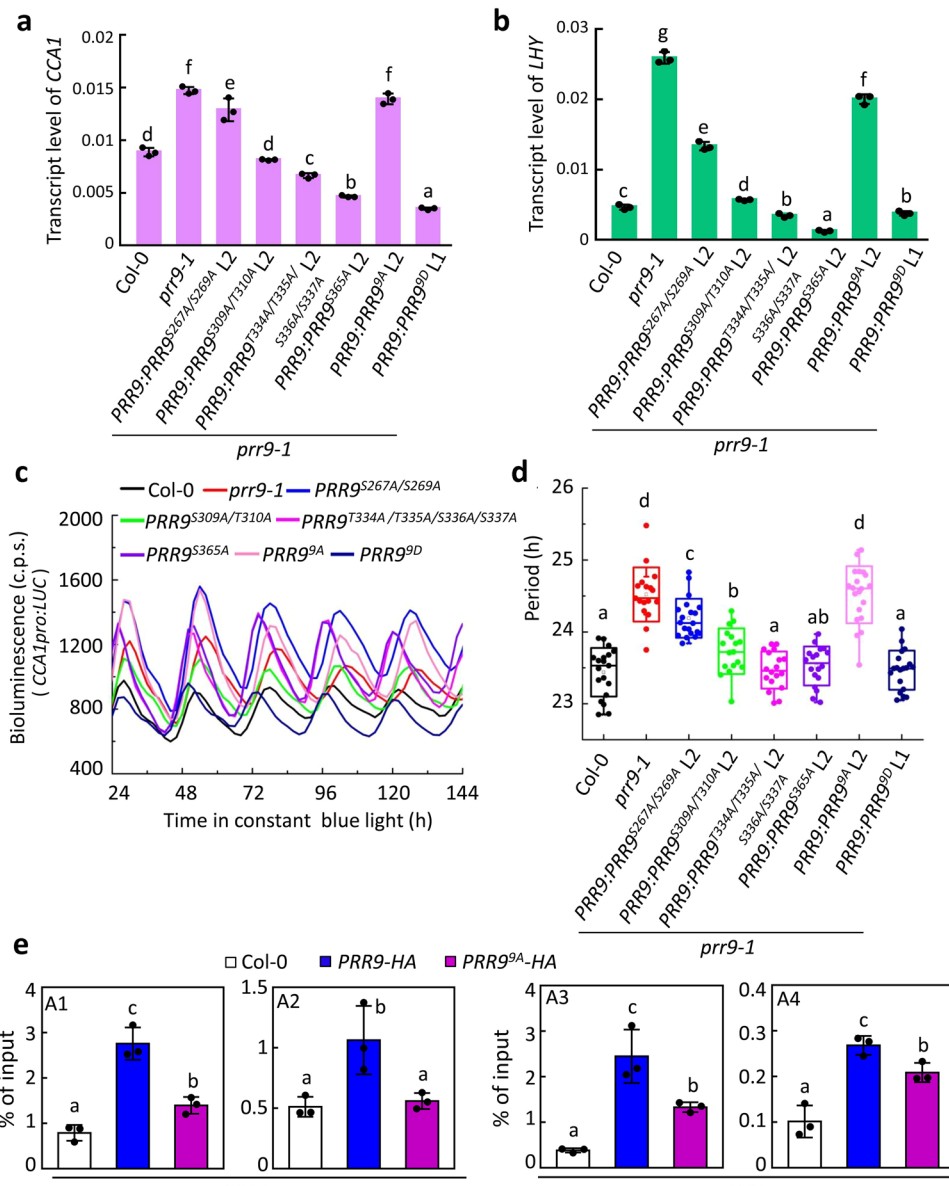

**Fig. 4 | Phosphorylation is necessary for circadian function of PRR9.**
**a**, **b** Transcriptional expression levels of *CCA1* (**a**) and *LHY* (**b**) in a variant of transgenic PRR9 phosphosites lines. 14-day-old seedlings of Col-0, *prr9-1*, lines of PRR9 de-phosphorylation or phosphorylation mimicking variants in *prr9-1* background grown in LD condition were harvested at ZT5 and tested by RT-qPCR. Data were mean ± s.d., *n* = 3, technical repeats. Lowercase letters indicate significant differences by one-way ANOVA followed by Fisher's LSD test. **c**, **d** Bioluminescence traces of *CCA1pro:LUC* (**c**) and the estimated free-running period (**d**) of seedlings as described in **a** under continuous blue light (40 μmol m⁻² s⁻¹) condition. Lowercase letters in **d** indicate significant differences by one-way ANOVA followed by Fisher's

LSD test. Data were mean ± s.d. Circles represent single data points; boxes: range of s.d., coef = 1; whisker: range of outlier, coef = 1.5; top, center, and below lines represent maximum, mean, and minimum values, respectively; "*x*": 1 and 99 percentile. **e** Relative binding of PRR9 and PRR9⁹ᴬ to *CCA1* and *LHY* promoters. Two weeks of Col-0, *CsVMVpro:PRR9-HA L2* and *CsVMVpro:PRR9⁹ᴬ-HA L3* seedlings entrained in LD cycles were harvested at ZT5 for ChIP assay with HA antibody. The enrichment of amplicons were analyzed by ChIP-qPCR. Lowercase letters in indicate significant differences among groups through one-way ANOVA followed by the Fisher's LSD test. Data were mean ± s.d., *n* = 3, technical repeats. Representative data of figure **a**−**e** from three independent experiments.

repress the higher expression of *CCA1* and *LHY* (Fig. 4a, b), even if its transcript and protein levels were higher than S309A/T310A transgenic line (Supplementary Fig. 4a, b). These results suggested that the 9 identified phosphosites were essential for PRR9 to act as a transcriptional repressor. Secondly, transgenic *PRR9⁹ᴰ* could fully revert the long circadian period phenotype of *prr9-1* mutant, while *PRR9⁹ᴬ* could not rescue the lengthened circadian period of *prr9-1* (Col-0 = 23.44 ± 0.07 h, *prr9-1* = 24.52 ± 0.09 h, *PRR9⁹ᴰ* L1 = 23.46 ± 0.06 h, *PRR9⁹ᴬ* L2 = 24.56 ± 0.65 h) (Fig. 4c, d). In addition, unlike other phosphosite mutations, S267A/S269A only partially rescued the circadian phenotype of *prr9-1* (*PRR9^{S267A/S269A}* L2 = 24.19 ± 0.06 h, *PRR9^{S309A/T310A}*

L2 = 23.73 ± 0.07 h, *PRR9^{T334A/T335A/S336A/S337A}* L2 = 23.46 ± 0.06 h, *PRR9^{S365A}* L2 = 23.52 ± 0.06 h) (Fig. 4c, d). The results further reinforce the idea that phosphorylation of PRR9 contributes to circadian rhythmicity regulation and phosphorylation of S267/S269 is important for PRR9 function. Since S267/S269 are close to the EAR motif, we asked if the phosphorylation status of PRR9 would affect its interaction with TPL. Surprisingly, we found a similar affinity between PRR9⁹ᴬ and PRR9⁹ᴰ with TPL (Supplementary Fig. 4e). In addition, we found that the phosphorylation status of PRR9 did not affect its interaction with CRY2 (Supplementary Fig. 4f). Given these unexpected observations, we then asked if PRR9 phosphorylation could affect its DNA binding

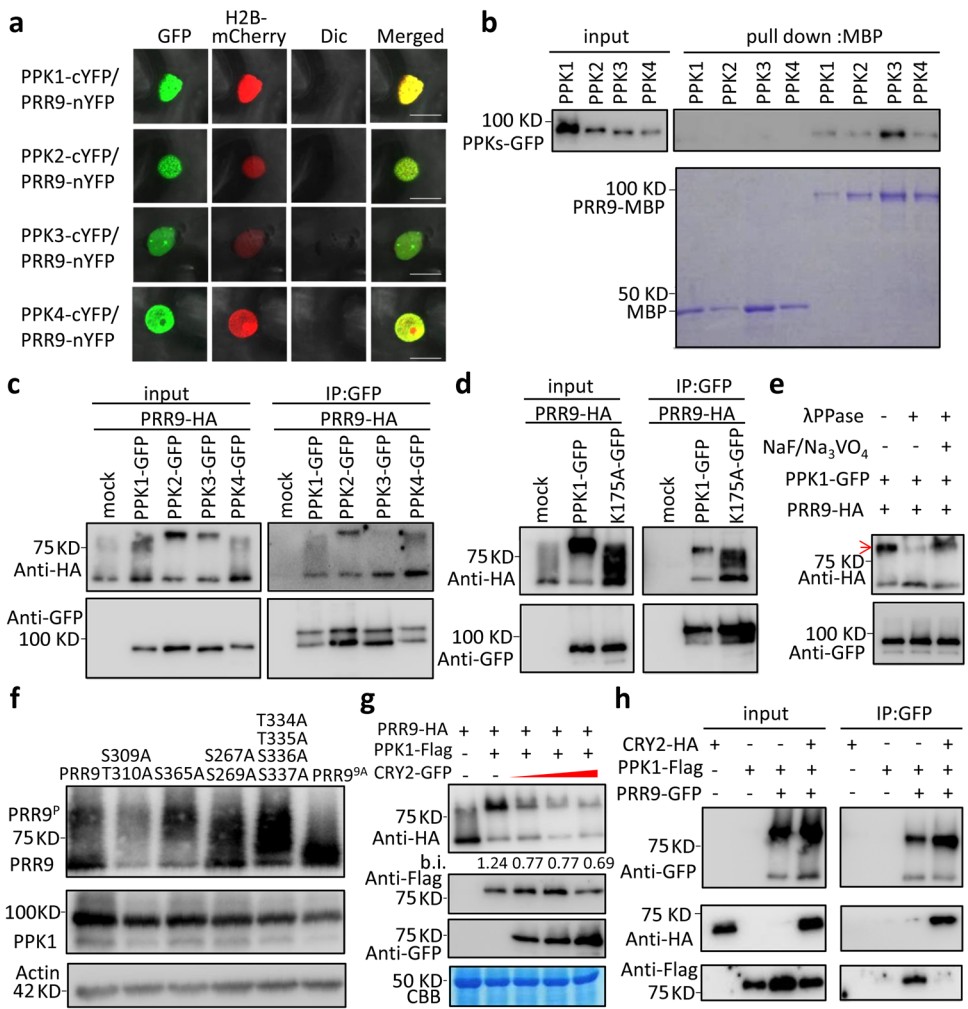

**Fig. 5 | CRY2 inhibits PPKs-promoted phosphorylation of PRR9. a** BiFC assay showing all PPKs (PPK1, PPK2, PPK3, and PPK4) interacted with PRR9 in the nucleus. PPKs-cYFP, were individually co-expressed with PRR9-nYFP or nYFP in *N. benthamiana* leaves. H2B-mCherry as a nucleus marker. Bars, 20 μm. (*n* = 12, 12, 10, 12 cells/2 leaves) **b** In vitro pull down of MBP or PRR9-MBP with PPKs-GFP protein. Pulled-down complexes were detected with GFP antibody, whereas MBP or PRR9-MBP were stained by CBB. **c** Co-IP assay showing PPKs-GFP interacted with PRR9-HA in *N. benthamiana*. The IPed (PPKs) and co-IPed signals (PRR9) were detected in immunoblots probed with GFP and HA antibodies, respectively. **d** Co-IP assay showing an interaction between PRR9-HA with PPK1 and PPK1^K175A-GFP. The IPed (PPK1 and PPK1^K175A) and co-IPed signals (PRR9) were detected in immunoblots probed with GFP and HA antibodies, respectively. **e** λPPase treatment of PRR9-HA with PPK1-GFP co-expressing *N. benthamiana* protein lysates. The red arrow

indicates phosphorylated PRR9. **f** Immunoblot showing GFP-tagged-PPK1 catalyzed phosphorylation of PRR9 and series of PRR9 mutants fused with HA tag in tobacco leaves. Total protein lysates were analyzed by immunoblot with GFP and HA antibodies, respectively. **g** Immunoblot showing PPK1-promoted phosphorylation of PRR9 was inhibited by CRY2. Red triangle indicates increased CRY2 concentration. b.i. stands for phosphorylated/unphosphorylated PRR9 band intensity ratio. PRR9-GFP, PPK1-Falg, and CRY2-HA were analyzed by immunoblot with GFP, Flag, and HA antibodies, respectively. CBB staining was used as a loading control. **h** Co-IP assay showing CRY2 interfered with PRR9/PPK1 interaction. The IPed (PRR9) and co-IPed signals (PPK1 and CRY2) were detected in immunoblots with GFP, Flag, and HA antibodies, respectively. For figure (**b–g**), similar results were observed from three biological repeats.

ability. The chIP-qPCR assay showed that the amplicons of A1, A2 on the *CCA1* promoter and A3, A4 on the *LHY* promoter were highly enriched in *PRR9-HA* but were significantly reduced in *PRR9^9A-HA* immunoprecipitates (Fig. 4e), suggesting that DNA binding ability of PRR9^9A was largely weakened compared to that of PRR9. Collectively, these results indicate that dynamic phosphorylation of PRR9 is critical for its DNA binding activity and a proper circadian clock function.

**CRY2 inhibits PPKs-mediated phosphorylation of PRR9**
Besides the identification of PRR9 phosphorylation sites, the IP-MS analysis also identified the in vivo interactome of PRR9. Indeed, PHOTOREGULATORY PROTEIN KINASEs (PPKs), including four members which were previously shown to catalyze CRY2 phosphorylation in blue light-dependent manner[51,52], were found to co-precipitate with PRR9 (Supplementary Fig. 5a). Thus, we reasoned that PPKs might

phosphorylate PRR9, which could be inhibited by CRY2 due to the competitive interaction among them. Evidently, the fluorescence signals were observed in the nuclei of epidermal cells of *N. benthamiana* when PRR9-nYFP co-expressed with any of the PPKs tagged by the C-terminus of YFP, but no signals were detected in the combination of either PPKs-cYFP with nYFP or cYFP or PRR9-nYFP alone, suggesting that PRR9 physically interacts with PPKs *in planta* (Fig. 5a and Supplementary Fig. 5b). Next, in vitro pull-down assay demonstrated that PPKs-GFP could be pulled down by PRR9-MBP but not by MBP negative control (Fig. 5b), indicating PRR9 directly interacts with PPKs in vitro. Finally, the interaction between PRR9 and PPKs were substantiated via co-IP assay by co-expressing HA-tagged PRR9 and GFP-tagged PPKs in *N. benthamiana* (Fig. 5c). Together, these results indicated that PPKs could physically interact with PRR9. Moreover, we noticed that unlike the reduced PRR9 phosphorylation by co-expressing with CRY2, a

slower-migrating band specifically shown up when PRR9 was co-expressed with PPKs, which might be the phosphorylated PRR9 band. Therefore, we examined the band shift of PRR9 by co-expressing PRR9 with PPK1$^{K175A}$, a nonfunctional allele of PPK1 without catalytic activity[51]. Interestingly, the interaction between PPK1$^{K175A}$ and PRR9 was comparable or even slightly stronger than that of PPK1 in the co-IP assay (Fig. 5d). However, PRR9 protein migrated much faster in the presence of PPK1$^{K175A}$ compared to PPK1 (Fig. 5d), suggesting that PPK1 catalytic activity was required for PRR9 phosphorylation. To further test this hypothesis, we treated the protein extracts of PRR9-HA and PPK1-GFP with λPPase. Clearly, the slower-migrating band of PRR9 disappeared after λPPase treatment (Fig. 5e). We then compared the band shift pattern of PRR9 phosphosite mutations by co-expressing with PPKs. Results showed that PPK1 and PPK4 display similar roles in catalyzing PRR9 phosphorylation, for the band shift promoted by PPK1 and PPK4 almost disappeared when they were co-expressed with PRR9$^{9A}$ compared to PRR9 and other phosphosite mutations (Fig. 5f and Supplementary Fig. 5e). While the phosphorylation pattern is slightly different from PPK2 and PPK3, as we observed PPK2 and PPK3 promoted specific band shift migrates slightly faster when co-expressed with PRR9$^{9A}$ and PRR9$^{S267A/S269A}$ compared to PRR9 and other phosphosite mutations (Supplementary Fig. 5c, d), suggesting PPK2 and PPK3 may recognize other unidentified residues besides S267 S269.

As CRY2 could interact with PPKs and inhibit PRR9 phosphorylation (Fig. 1e, f), we then investigated if CRY2-induced blocking of PRR9 phosphorylation was caused by its competitive interaction with PPKs. The results showed that the upper phosphorylated band of PRR9-HA protein was present when co-expressing with PPK1-FLAG. Importantly, the upper bands of PRR9-HA gradually decreased with the increase of CRY2 protein level (Fig. 5g), suggesting that PPK1-mediated PRR9 phosphorylation was blocked by CRY2. Moreover, the competitive co-IP assay demonstrated that the interaction between PRR9 and PPK1 was significantly diminished by CRY2 (Fig. 5h), further suggesting that CRY2 impeded PPKs–PRR9 interactions, which consequently inhibited PPKs-mediated PRR9 phosphorylation of PRR9.

## PRR9 is required for PPKs to regulate the circadian clock

The circadian phenotypes of PPK single mutants have been reported[53], that *ppk1* has no circadian phenotype, while *ppk2*, *ppk3*, and *ppk4* single mutant all exhibits long free-running period. Besides, the circadian period of *ppk1 ppk2*, *ppk1 ppk4*, and *ppk1 ppk2 ppk4* double or triple mutants are similar to wild type, meaning *ppk1* can restore the circadian phenotype of other PPKs to the wild type. Hence, we choose the evolutionary closed PPK2 and PPK3 to analyze the circadian phenotype. Firstly, we found that PRR9 protein level in *ppk2 ppk3* double mutant was much higher than that in Col-0 from ZT1 to ZT9 (Fig. 6a, b), while the transcript level of *PRR9* in *ppk2 ppk3* mutant was comparable to that in the Col-0 (Fig. 6c). We then examined the expression of *CCA1* and *LHY*, the two known targets of PRR9 within the core TTFLs. As determined by RT-qPCR analysis, no significant reduction of *CCA1* and *LHY* transcript levels was found in *ppk2 ppk3* mutant (Fig. 6d, e), consistent with the notion that PRR9 phosphorylation is required for its transcriptional repression activity. To determine if PRR9 is genetically required for the regulation of the circadian clock modulated by PPKs, we crossed *prr9-1* mutant harboring *CCA1:LUC* as a circadian reporter with *ppk2 ppk3* double mutant. The circadian phenotypes of *prr9-1 ppk2 ppk3* triple mutant were investigated in both constant blue and constant red light conditions. We found that *prr9-1* displayed a longer circadian period than *ppk2 ppk3* in cB. Nevertheless, the circadian period of *prr9-1 ppk2 ppk3* triple mutant was not additively longer in cB (Col-0 = 23.69 ± 0.07 h, *prr9-1* = 24.64 ± 0.09 h, *ppk2 ppk3* = 24.02 ± 0.08 h, *ppk2 ppk3 prr9-1* = 24.17 ± 0.05 h). Intriguingly, *ppk2 ppk3* had a much longer circadian period than *prr9-1* in cR, while the circadian period of *prr9-1 ppk2 ppk3* triple mutant was comparable to that of *ppk2 ppk3* double mutant in cR (Col-0 = 23.14 ± 0.05 h, *prr9-*

*1* = 23.47 ± 0.07 h, *ppk2 ppk3* = 24.09 ± 0.11 h, *ppk2 ppk3 prr9-1* = 23.96 ± 0.08 h) (Fig. 6f, g). Given that PPKs interact with other circadian components as previously reported, such as ELF3[53] and CCA1[54], we proposed that the regulation of the circadian clock by PPKs at least in part requires PRR9 in cB but is independent of PRR9 in cR. Together, the results suggest that PRR9 works downstream of PPKs under blue light, but not red light.

## PRR9 acts downstream of CRY2 in a circadian clock module

Given the effect of CRY2 on the circadian period was pertinent to blue light intensity, therefore we analyzed the fluence response curve (FRC) of Col-0, *prr9-1*, *cry2*, and *cry2 prr9-1* seedlings carrying *CCA1pro:LUC* circadian reporter in constant blue light. The periods of circadian rhythm of *CCA1pro:LUC* in *cry2* or *prr9-1* mutant were longer than those of Col-0 regardless of the blue light intensity tested (Fig. 7a and Supplementary Table 1). Importantly, the circadian period of the *cry2 prr9-1* double mutant resembled the *prr9-1* single mutant, indicating that PRR9 acted genetically downstream of CRY2 in circadian period modulation in blue light (Fig. 7a and Supplementary Table 1). Moreover, free-running periods of *cry2 PRR9ox-1* seedlings were akin to *PRR9ox-1*, especially under intermediate and high blue light, further reinforcing that PRR9 was a major downstream clock component to convey CRY2-mediated blue light signaling to the clock. Noticeably, the free-running periods of *CCA1pro:LUC* in *cry2 PRR9ox-1* were modestly shorter than *PRR9ox-1* in low blue light (Fig. 7b and Supplementary Table 2), suggesting that the suppression of PRR9 by CRY2 was aggravated in limiting blue light. On the other hand, the free-running period of *cry1 prr9-1* double mutants was similar to either *cry1* or *prr9-1* single mutant in constant red light (Col-0 = 23.29 ± 0.04 h, *prr9-1* = 23.69 ± 0.08 h, *cry1* = 23.92 ± 0.07 h, *cry1 prr9-1* = 23.77 ± 0.1 h), while was much longer than either *cry1* or *prr9-1* single mutant in constant blue light (Col-0 = 23.32 ± 0.08 h, *prr9-1* = 24.44 ± 0.05 h, *cry1* = 24.83 ± 0.09 h, *cry1 prr9-1* = 25.46 ± 0.19 h), indicating that CRY1 and PRR9 act additively (Fig. 7c, d and Supplementary Fig. 6a, b). Together, our results demonstrated that PRR9 acts downstream of CRY2, but CRY1 has a distinct mechanism than CRY2 in mediating blue light input into the circadian clock.

Collectively, we propose a working model, in which blue light triggers rapid phosphorylation of CRY2 by PPKs that facilitates subsequent degradation of CRY2 proportional to blue light intensity. CRY2 degradation would then cause the release of PRR9 from the CRY2-PRR9 complex for PPKs to access and phosphorylate PRR9. Phosphorylated PRR9 could then recruit co-repressor TPL to effectively bind and repress target genes, leading to a shorter free-running period (Fig. 7e, upper panel). By contrast, photoexcited CRY2 is not subjected to rapid turnover in limiting blue light, resulting in a more stable CRY2-PRR9 complex, hence blocking PRR9-PPKs and PRR9-TPL interactions, which diminishes the transcriptional repression of PRR9 on the target genes, ultimately causing a relatively longer free-running period (Fig. 7f, lower panel).

## Discussion

CRY1 and CRY2 have been manifested to indirectly mediate the regulation of circadian through interacting with PIF4 and PIF5 proteins in a blue light-dependent manner, while PIF proteins are involved in fine-tuning of circadian clock[39,55–59]. Recently, CRYs were also shown to interact with METTL3/14 type m⁶A writers and the light-induced phase separation of CRYs can modulate the m⁶A writers activity on the mRNAs of *CCA1* and other clock components[38]. These studies tend to think CRY1 and CRY2 act in the same mechanism to transmit blue light signaling, leaving their specific light-dependent stability property out of consideration[40,60]. Based on our current results, CRY2 mediates blue light signaling into the circadian clock by physically interacting with PRR9, which enables *Arabidopsis* clock to sense the changing light intensity. The effects of blue light on CRY2 stability, together with the

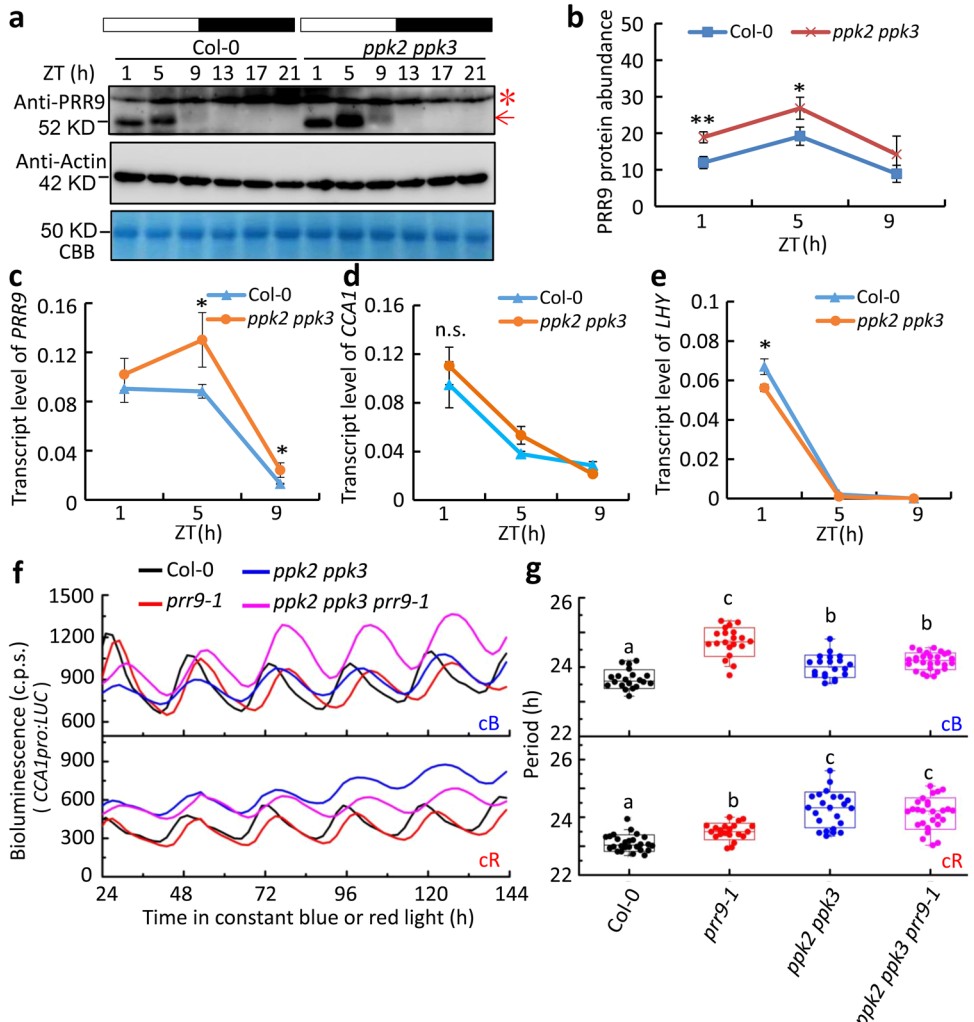

**Fig. 6 | Phosphorylation is critical for both protein turnover and repressive activity of PRR9. a** Temporal expression pattern of PRR9 protein in Col-0 and *ppk2 ppk3* double mutants. Seedlings were grown in LD condition for 14 days and harvested at indicated time points. Total protein lysates were analyzed by immunoblot probed with PRR9 antibody indicated by a red arrow. The asterisk stands for nonspecific band and the arrow indicates endogenous PRR9. Actin and CBB staining were used as loading controls. **b** Quantitative analysis of PRR9 protein abundance as shown in **a**, normalized with Actin. Data were means ± s.d. from three biological replicates. **\*\****p* < 0.01, **\****p* < 0.05 by two-tailed student's *t*-test. **c−e** Temporal transcript levels of *PRR9* (**c**), *CCA1* (**d**), and *LHY* (**e**) from seedlings as shown in **a** grown in LD conditions, determined by RT-qPCR. Data were mean ± s.d.,

*n* = 3, technical repeats. Similar results were observed from three independent biological repeats. **\****p* < 0.05, n.s., indicates no significant by two-tailed student's *t*-test. **f, g** Bioluminescence traces of *CCA1pro:LUC* (**f**) and the estimated free-running period (**g**) for Col-0, *prr9-1*, *ppk2 ppk3*, and *ppk2 ppk3 prr9-1* seedlings under continuous red (*n* = 20, 20, 20, 27, respectively) or blue light (*n* = 26, 21, 23, 27 respectively) (40 μmol m⁻² s⁻¹) conditions. Data were mean ± s.d., and lowercase letters in **g** indicate significant differences by one-way ANOVA followed by the Fisher's LSD test. Circles represent single data points; boxes: range of s.d., coef = 1; whisker: range of outlier, coef = 1.5; top, center, and below lines represent maximum, mean, and minimum values, respectively; "x": 1 and 99 percentile. Data from three independent biological repeats.

interaction and inhibition of CRY2 on PRR9 activity, ultimately determine the circadian speed in *Arabidopsis*. Nonetheless, our genetical evidence unambiguously supports that PRR9 and CRY2, but not CRY1, act in the same pathway to modulate the circadian period (Fig. 7). Thus, we propose that there are intricate downstream networks, some shared by CRY1 and CRY2, others are distinctive, in transmitting CRYs-mediated light signals to the circadian clock.

Although post-translational regulation is pivotal for plant circadian clock function, very little is known for the kinases involved. In this study, using mass spectrometry followed by PRR9-GFP immunoprecipitation, we discovered that PPKs are the kinases that phosphorylate PRR9. Nevertheless, PPKs were shown to be co-precipitated with ELF3 and ELF4 in a tandem affinity purification followed by mass spectrometry analysis[53]. In addition, PPKs also interact and phosphorylate CRY2 directly[51]. Hence, it is conceivable that PPKs is important for fine-tuning the circadian clock by recognizing multiple targets such as

PRR9, CRY2, and ELF3. Therefore, our findings revealed that PPKs, the conserved kinases from unicellular algae to higher plants, are players in modulating the circadian clock, at least through mediating PRR9 phosphorylation and enhancing PRR9 DNA binding ability. Besides, We observed differential accumulation of phosphorylated PRR9 between PRR9 variants when they were co-expressed with PPKs, suggesting some phosphosites is associated with phosphorylated PRR9 protein stability, while the underlying mechanisms wait for further exploring. More efforts are warranted to identify the PPKs phospho-codes on clock components, which would be essential for a deeper understanding of the roles of PPKs in circadian pacemaking, plant growth, and flowering time control.

Plant photoreceptors CRYs and phytochromes have long been proposed to entrain and reset the plant circadian clock by transmitting light signals[35,37]. Here we revealed the role of CRY2 in conveying light input to the clock through a direct association and negative regulation

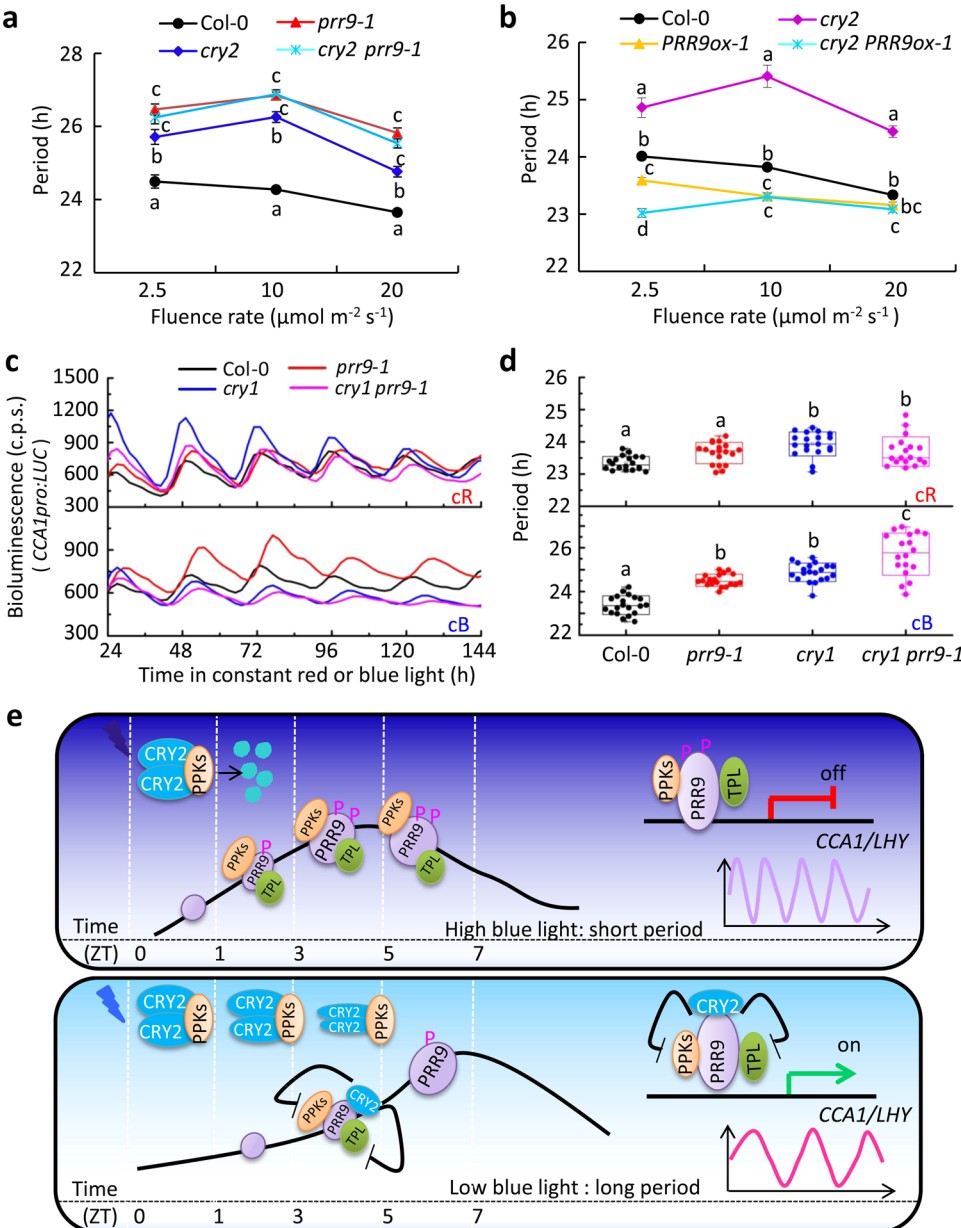

**Fig. 7 | PRR9 genetically acts downstream of CRY2 in regulating the circadian period. a, b** Fluence response curve showing free-running period length of *CCA1-pro:LUC* in Col-0, *prr9-1*, *cry2*, and *cry2 prr9-1* plants (**a**) (2.5 μmol m⁻² s⁻¹: *n* = 19, 16, 16, 18; 10 μmol m⁻² s⁻¹: *n* = 17, 19, 18, 19; 20 μmol m⁻² s⁻¹: *n* = 20, 19, 20,19, respectively) or Col-0, *cry2*, *PRR9ox-1*, and *cry2 PRR9ox-1* plants (2.5 μmol m⁻² s⁻¹: *n* = 17, 17, 17, 15; 10 μmol m⁻² s⁻¹: *n* = 17, 17, 18, 15; 20 μmol m⁻² s⁻¹: *n* = 18, 18,18,19, respectively) under constant blue light at sequential fluent rates. Data represent mean ± s.e.m. Lowercase letters in **a** and **b** indicate significant differences, different genotypes at each light intensity were respectively analyzed through one-way ANOVA followed by the Fisher's LSD test. **c, d** Bioluminescence traces of *CCA1pro:LUC* (**c**) and the estimated free-running period (**d**) for Col-0, *prr9-1*, *cry1*, and *cry1 prr9-1* seedlings under continuous red (*n* = 20, 19, 19, 18, respectively) or blue light (*n* = 20, 19, 20, 18, respectively) (40 μmol m⁻² s⁻¹) condition. Data were mean ± s.d., and lowercase letters in **d** indicate significant differences by one-way ANOVA followed by the Fisher's LSD test. Circles represent single data points; boxes: range of s.d., coef = 1; whisker: range of outlier, coef = 1.5; top, center, and below lines represent maximum, mean, and minimum values, respectively; "*x*": 1 and 99 percentile. **e** A proposed working model for the interaction of CRY2 and PRR9 mediated environmental light intensity information to the circadian clock. High blue light promotes phosphorylation and degradation of CRY2, which enables PRR9 recruiting PPKs for phosphorylation and interacting with TPL to form transcription repressive complex to inhibit the expression of *CCA1* etc., leading to a short period (upper panel). While in the limiting blue light, CRY2 sequestrates PRR9 function through competing its interaction with TPL and PPKs, causing high expression of *CCA1* etc., leading to a long period (lower panel).

of clock protein PRR9. This resembles a mechanism in *Drosophila*, in which light exposed CRY binds to core clock component TIMELESS (TIM) and triggers TIM turnover to release PERIOD–TIM mediated transcriptional repression for synchronizing the circadian clock with light–dark cycles[29,60,61]. These findings suggest that the mechanism by which photoreceptors directly interact with clock components and post-translationally regulate circadian clock entrainment is conserved across different kingdoms. Furthermore, as a few plant photoreceptors such as CRYs and PHYs also play necessary roles in sensing temperature[62–64], temperature information might also be transmitted to the circadian clock via a similar protein-protein interacting network. Hence, the intricate interacting network between photoreceptors and clock components could play a much more extensive role in various biological processes that awaits further elucidation.

## Methods

All *Arabidopsis* single mutants used in this work are of Columbia-0 (Col-0) accession, including *prr9-1* (SALK_007551)[65]; *ppk2-1* (SALK_026482), *ppk3-2* (SALK_000758)[51,52]; *cry1* (*hy4-2.23*) and *cry2-1*[35]. *cry1 prr9-1*, *cry2 prr9-1* and *ppk2 ppk3* double mutants were obtained by the genetic cross of the respective single mutant. The *ppk2 ppk3 prr9-1* triple mutant was prepared by a genetic cross of *ppk2 ppk3* and *prr9-1*. The homozygous double or triple mutants were identified by genomic PCR screening from F2 segregating population. Genomic PCR scoring primers were listed in Supplementary Table 3. *CsVMV:PRR9-HA*, *CsVMV:PRR9-GFP*, or *35S:GFP-CRY2* overexpression transgenic lines were generated in Col-0 *CCA1pro:LUC* background, while *PRR9* phospho-mutant transgenic lines are in *prr9-1 CCA1pro:LUC* background. *cry2 PRR9ox-1* were generated by the genetic cross of *cry2* and *CsVMV:PRR9-HA L1 (PRR9ox-1)*, and homozygous seedlings were screened on Murashige and Skoog medium containing hygromycin B. All transgenic lines used in the work were generated by *Agrobacterium tumefaciens*-mediated transformation method. Stable and hereditary F3 progeny were used for biochemical analysis.

### Vector construction

All the coding sequences in this work were respectively amplified by PCR from cDNA. For *in planta* expression, *PRR9*, *CRY1* and *CRY2* cDNA without stop codon were respectively subcloned between *Kpn* I and *Xma* I sites of *pCsVMV:HA-1300* binary vector to generate C-terminal HA fusions. *PRR9*, *PRR9N*, *PRR9C*, *PRR9D*, *CRY2* and *PPKs* cDNA without stop codon were respectively subcloned into *Kpn* I and *Xma* I sites of *pCsVMV:GFP-1300* binary vector to generate C-terminal GFP fusions. Point mutation of *PPK1^{K175A}* was generated by overlapping PCR through site-directed mutation in the primer, then subcloned into *Kpn* I and *Xma* I sites of *pCsVMV:GFP-1300*. To generate *35S:PPK1-Flag* constructs, the *PPK1* fragment was subcloned into *Sal* I and *Bam*H I sites of *pCM1307* vector. To generate *PRR9* mutant variants driven by a native promoter or *CsVMV* promoter, *PRR9* mutant fragments were firstly obtained by overlapping PCR through site-directed mutation in the primer and then subcloned into *pPRR9:HA-1300* vector or *pCsVMV:HA/GFP-1300* vector, respectively. To generate *35S:GFP-CRY2*, *35S:GFP-PHR*, and *35S:GFP-CCE* constructs, *CRY2* and its fragments were firstly subcloned into *Kpn* I and *Xho* I sites of the *pENTR2B* vector to get entry clone, and then subcloned into *35S:GFP* gateway destination vector via LR reaction. *35S: PRR9-nYFP* was generated by subcloning *PRR9* into *Pac* I and Spe I restriction sites of the *2YN_pBI* vector, while *35S:CRY1-cYFP*,*35S:CRY2-cYFP*, and *35S:PPKs-cYFP* were generated by subcloning respective fragments into *Pac* I and Spe I restriction sites of *2YC_pBI* vector. *PRR9-NanoLucN* was generated by overlapping PRR9 with NanoLucN and subcloning the PRR9-NanoLucN fragment into *Kpn* I and *Xma* I restriction sites of the *pCsVMV-1300* vector. *CRY1-NanoLucC* and *CRY2-NanoLucC* were obtained by subcloning *CRY1* and *CRY2* fragment into *Kpn* I and *Xma* I restriction sites of the *pCsVMV:NanoLucC-1300* vector. To purify PRR9-MBP protein, PRR9 was firstly subcloned into sites of the *pMAL2C* vector to generate N-terminal fusions (MBP-PRR9) and then transformed into *E.coli* strain BL21. Constructs of TPL-Flag, *pCCApro:LUC* and *pCsVMVpro:LUC* are from previously reported[66]. Unless otherwise indicated, all the vector constructs were carried out by One Step Cloning Kit (Vazyme) according to the manufacturer's protocol and confirmed by Sanger-sequence. Vector construction primers were listed in Supplementary Table 3.

### Bioluminescence assay

For bioluminescence assay, seeds were firstly sterilized and grown on MS medium containing 3% sucrose, then kept at 4 °C for 3 days, followed by entrainment in an incubator for 7 days with system parameter setting as 22 °C, 100 µmol m$^{-2}$ s$^{-1}$,12-h light/12-h dark cycles. Before fluorescence signals acquisition, 7-day-old seedlings were respectively moved to a 9 cm × 9 cm MS + 3% sucrose culture dish with

1 cm interval, followed by spraying with working buffer (20 µL D-Luciferin and 1 µL 20% Triton X-100 in 1 mL ddH$_2$O) on seedlings. Bioluminescence signals are gathered using a CCD camera (LN/1300-EB/1, Princeton Instruments) in constant blue or red-light conditions for 7 days with 2-h intervals for exposure and 30 min dark for signal acquisition. Light intensity was set as 40, 20, 10, or 2.5 µmol m$^{-2}$ s$^{-1}$ by overlapping 0, 1, 2, or 4 layers of filters with 50% light transmittance on the culture dish. Real-time bioluminescence raw data were measured using MetaMorph Microscopy Automation and Image Analysis Software, and the exported bioluminescence intensity values were analyzed by OriginPro 8.5.1 software to make bioluminescence trace and further analyzed by the Biological Rhythms Analysis Software System (BRASS v2.14, available from www.amillar.org) to calculate period lengths and relative amplitude from individual traces using the Fourier transform-nonlinear least-squares suite of programs with a time window from 24 to 144 h.

### Bimolecular fluorescence complementation assay

For the BiFC assay, *agrobacteria* containing plasmids expressing fusion protein tagged with nYFP, cYFP, and plasmids expressing H2B-mCherry, which was used as a nuclear marker, were co-infiltrated into *N. benthamiana* leaves in a ratio of 5:5:1 for 3 days. The infiltrated leaves were cut into small pieces and examined under Olympus FV1000MPE confocal microscope with signals excited by 488 and 563 nm laser sets.

### Split nano-luciferase complementation assay

For split Nano-LUC assay, *Agrobacteria* co-expressing PRR9-NanoLucN with NanoLucC, CRY1-NanoLucC or CRY2-NanoLucC were respectively transfected into *N. benthamiana* leaves for 3 days. The substrate solution (1 µL 10 mM furimazine and 1 µL 20% Triton X-100 in 1 mL ddH$_2$O) was injected into the infiltrated leaves and kept at room temperature for 5 min before collecting bioluminescence by a CCD camera (LN/1300-EB/1, Princeton Instruments).

### Protein extraction and co-immunoprecipitation

For protein expression assay in *N. benthamiana* system, *Agrobacteria* containing purposed proteins were co-infiltrated as indicated into tobacco leaves and cultured for 3 days in a chamber at 25 °C with 16-h light/8-h dark conditions. Leaves were harvested and ground in liquid nitrogen into powder. As for co-IP assay in *N. benthamiana* system, 0.5% NP-40 in protein extraction buffer (10 mM Tris-HCl pH 8.0, 150 mM NaCl, 0.5 mM EDTA pH 8.0, 0.5% NP-40, 1 mM DTT, 1 mM PMSF, 5 µg/ml Chymostatin, 5 µg/ml Leupeptin, 5 µg/ml Pepstatin, 5 µg/ml Aprotinin, 5 µg/ml Antipain, 2 mM Na$_3$VO$_4$, 2 Mm NaF, 50 µM MG132, 50 µM MG115, and 50 µM ALLN) was reduced to 0.2%. The homogenate was centrifuged at 18,000×*g* for 8 min and the extracted supernatant was incubated with freshly cleared GFP Trap beads (ChromoTek) (10 µl beads slurry every sample) at 4 °C with gentle agitation for 1 h. The immune complex was washed four times using ice-cold washing buffer (10 mM Tris-HCl pH 8.0, 150 mM NaCl, 0.5 mM EDTA pH 8.0, 0.2% NP-40, 1 mM DTT, 1 mM PMSF) by the magnetic stand and eluted from beads by 6 × SDS loading buffer followed by heating at 55 °C for 1 min. The mixtures were then transferred to the western blot procedure and probed with respective antibodies. For Co-IP in *Arabidopsis*, the protein extraction method was similar except the following details: protein inhibitors were double added, 30 µl beads slurry per sample and the immune complex was washed two times by washing buffer.

### Lambda PPase treatment

For the λPPase treatment assay, tobacco leaves tissues were harvested and ground in liquid nitrogen into powder. Total proteins were extracted by suspending tissue power with an equal volume of lambda phosphatase buffer from NEB company (1x PMP buffer, 0.5% Triton X-100, 0.4% NP-40, 2 mM MnCl$_2$, 50 µM proteasome inhibitors, and 1 mM PMSF), vortex and centrifuge at 4 °C with 18,000×*g* for 8 min. The

supernatant were equally divided into two tubes. Half were treated with λPPase only and the half were treated with λPPase and phosphatase inhibitors (NaF and $Na_3VO_4$). Both were kept. at 30 °C for 5 min. The reaction was stopped by adding phosphatase inhibitors and 6 x SDS loading buffer at room temperature for 2 min. The samples were then transferred to electrophoresis with 8% special gels.

## Protein pull-down assay

Tobacco leaves or *Arabidopsis* seedlings were ground in liquid nitrogen and homogenized with extraction buffer. After centrifuging at 4 °C for 8 min at 18,000×g, the protein extraction supernatant were aliquoted into two tubes with 20 μl recombinant PRR9-MBP or MBP resin and incubated at 4 °C with gentle agitation for 2 h. The protein-beads complex were then respectively washed by washing buffer 1 (50 mM Tris-HCl pH 8.0, 150 mM NaCl, 1 mM EDTA pH 8.0, 0.5% NP-40, 0.5% Triton X-100, 3 mM DTT, 1 mM PMSF) for 10 min, washing buffer 2 (with only 0.1% NP-40 and 0.1% Triton X-100) for 5 min and washing buffer 3 (without NP-40 and Triton X-100) for 5 min. Finally, proteins were eluted with 2 X SDS-PAGE buffer prior to electrophoresis. MBP and PRR9-MBP proteins were stained by Coomassie Brilliant Blue R250. Pulled-down proteins were detected with GFP antibody.

## Chromatin immunoprecipitation

Seedlings used for ChIP assay were grown on MS medium containing 3% sucrose in a chamber in LD condition (12 h light/12 h dark) for 14 days, then cross-linked with 1% formaldehyde at ZT4. ChIP experiments were performed as described[67]. Materials were sequentially ground, extracted, sonicated (30-s ON/30-s OFF for six times, each with 5 min), precleared with 40 μl slurry of salmon sperm sheared DNA/protein A agarose beads (16–157, Millipore, Billerica, USA) and bound with corresponding antibodies. For Col-0, *CsVMVpro:PRR9-HA* and *CsVMVpro:PRR9 9A-HA* transgenic materials, 6 μl ChIP grade HA antibody (ab9110, Abcam, Cambridge, USA) was used in each sample. For Col-0, *prr9-1*, *cry2*, and *cry2 prr9-1* materials, 4 μl ChIP grade H3 and H3K9ac antibody (Merck-Millipore) were used in each sample. The immunoprecipitated chromatin complexes bound to beads were washed with low salt, high salt, LiCl wash buffer for one time and TE buffer for two times sequentially. Finally, chromatin complexes were released from the beads with 200 μl elution buffer (1% SDS, 0.1 M $NaHCO_3$) and reverse cross-linked by adding 20 μl 5 M NaCl at 65 °C overnight. Purified DNA IP and input sample were diluted ten times with $ddH_2O$ and analyzed by qPCR. Mean values of $2^{\Delta(Ct(IP)-Ct(input))}$ were calculated and normalized to Actin. Primers used for ChIP-qPCR were shown in Supplementary Table 3.

## Immunoprecipitation-mass spectrometry

Two-week-old *PRR9:PRR9-GFP* transgenic seedlings[25] grown on MS medium containing 3% sucrose in LD cycles (12 h light/12 h dark) were transferred at ZT0 and exposure to 40 μmol m$^{-2}$ s$^{-1}$ blue light for 5 h. Plant tissues (3 g) were harvested, ground in liquid nitrogen and homogenized in 3 ml clod IP buffer. The protein lysates were vortexed violently and filtered by four layers of 0.2 μm Miracloth (EMD Millipore Corp) through a centrifuge at 15,000×g for 10 min at 4 °C. The supernatant were added with precleared GFP Trap beads (30 μl beads slurry every sample) and incubated at 4 °C with gentle agitation for 1 h. Then, protein-beads complex were washed with 1 ml washing buffer 1 (10 mM Tris/Cl pH 8.0; 150 Mm NaCl; 0.5 mM EDTA; 0.5% NP-40) for four times and 1 ml washing buffer 2 (10 mM Tris/Cl pH 8.0, 150 Mm NaCl, 0.5 mM EDTA) for two times and 1 ml 50% acetonitrile for four times by using magnetic stander. Finally, the protein-beads complex were resuspended and digested by elution buffer 1 (50 mM Tris/Cl pH 7.5; 2 M urea; 5 μg/ml sequencing grade modified trypsin; 1 mM DTT) at 30 °C for 30 min. After centrifuging at 4 °C, 800×g for 2 min, the supernatant were transferred to a new vial. The remaining beads were further resuspended by elution buffer 2 (50 mM Tris/Cl pH 7.5; 2 M

urea; 5 mM iodoacetamide) twice. The total supernatant from three times of elution were kept at 30 °C overnight for digestion. The digested samples (1–2 μg protein) were submitted to Thermo Scientific Orbitrap Fusion Lumos Tribrid mass spectrometer coupled with a chromatography system, namely a 20 cm EASY-Spray C18 LC column (1.9 μm particle size) with a 2–40% acetonitrile gradient over 75 min at a flowrate of 500 nL/min to separate peptides. Orbitrap Fusion Lumos Tribrid platform was used for further peptide analysis. Briefly, typical analysis at 120,000 resolving power survey scan, AGC 4e5 followed by MS/MS analysis at 1.6 m/z isolation with the quadrupole, HCD 32% collision energy, then fragment ions analysis in the orbitrap analyzer with resolution 15,000. For MS/MS analysis, the maximum injection time was appointed as 72 ms with an AGC target of 1e4, and charge states 1 or >7 were excluded. Thermo Scientific™ Proteome Discoverer™ 2.4 software was used for data analysis. Precursor mass tolerance and fragment mass tolerance was set as 10 ppm and 0.02 Da, respectively. Carbamidomethylation (+57.021 Da) was used as a fixed modification, while methionine oxidation (+15.996 Da), phosphorylation (+79.966 Da, T, Y, S), and acetylation on protein N-terminal (+170 Da) were set as variable modifications. Data were searched against a Uniprot-*Arabidopsis thaliana* database with a 1% FDR criteria to filter the results. The mass spectrometry proteomics data have been deposited to the ProteomeXchange Consortium via the PRIDE[68] partner repository with the dataset identifier PXD035252.

## Transient expression assay in *N. benthamiana*

*CCA1pro:LUC* and *CsVMVpro:LUC* were used as reporters, *GFP, PRR9-GFP*, and *CRY2-GFP* were used as effectors. *Agrobacteria* suspension carrying reporter and effector was infiltrated into *N. benthamiana* leaves for 2 days by a syringe infiltration method. The complete leaf was immersed into 1 ml $ddH_2O$ with 20 μl luciferase and 0.01% Triton X-100 for 3 min before collecting the bioluminescence signal with the CCD camera. The averaged bioluminescence intensity of LUC signals were measured and quantified by Metamorph software.

## *Arabidopsis* protoplasts transient expression assay

*Arabidopsis* mesophyll protoplasts were isolated from well-expanded fourth, fifth, and sixth leaves of Col-0 plants[22]. Briefly, immersing 10–15 leaves with abaxial epidermis spread by adhesive tape into enzyme solution (0.4 M mannitol, 20 mM KCl, 20 mM MES, 10 mM $CaCl_2$, 1.2% w/v cellulose R10, 0.4% w/v macerozyme R10, 0.1% BSA), then incubate in the dark on the rotator shaker with slowly stirring at room temperature. Spin down protoplast with 100×g at 4 °C. Resuspend and wash protoplasts twice with W5 buffer (154 mM NaCl, 125 mM $CaCl_2$, 5 mM KCl, 1.5 mM MES, and 5 mM Glucose), then kept on rice for 30 min. Centrifuge with 100 × g at 4 °C for 5 min, remove supernatant and resuspend in MaMg buffer (0.4 M Mannitol, 15 mM $MgCl_2$, 4 mM MES). For each transformation, proportional 5 μg effector and 5 μg reporter plasmid was mixed with 230 μl protoplasts and 230 μl PEG solution (40% w/v PEG4000, 0.2 M Mannitol, and 0.1 M $Ca(NO_3)_2$), inverting eight times and incubate 8 min at room temperature. Transfected protoplasts were washed with 1 ml W5 buffer, then incubated overnight at 22 °C. Transcriptional activity was measured in next morning by detecting the LUC/REN ratio in cell lysates using a Dual-Luciferase Reporter Assay System (Promega).

## Quantitative real-time PCR gene expression analysis

*Arabidopsis* seedlings grown on MS medium with 3% sucrose were cultured in an incubator for 10 days, then transferred to red or blue light and harvested at the indicated time. *Zeitgeber* time (ZT) refers to the experimental LD cycle with ZT0 and ZT12 corresponding to lights on and off, respectively Total RNA was extracted using TRIzol Reagent (Life Technologies) as manual instruction described. About 1 μg RNA was used for reverse transcription by PrimeScript RT reagent Kit with gDNA Eraser (Takara). Quantitative real-time RT-PCR (qRT-PCR) was

performed by using SYBR Green Real-Time PCR Master Mix (TOYOBO, Osaka, Japan) according to the manufacturer's instructions on an Applied Biosystems™ QuantStudio 3 instrument (Applied Biosystems). Gene expression was normalized by *ACTIN2* expression. Mean values of $2^{-\Delta CT}$ were calculated from three biological replicates and three technical replicates were performed. Sequences of primers used for qPCR were listed in Supplementary Table 3.

### Quantification and statistical analysis

Data for quantification analysis were presented as mean ± s.e.m (standard error of mean) or s.d. (standard deviation) as indicated in figure legends. Statistical analyses were performed using SPSS software (https://www.ibm.com/products/spss-statistics) through one-way ANOVA followed by Fisher's LSD test or using EXCEL through two-sided student' *t*-test. In all graphs, letters indicate statistical significance ($p < 0.05$), ***$p < 0.001$, **$p < 0.01$, *$p < 0.05$, n.s., no significant. An exact *p* value of statistical tests were provided in the Source Data file.

### Reporting summary

Further information on research design is available in the Nature Research Reporting Summary linked to this article.

## Data availability

Other materials of this study are available from the corresponding author upon reasonable request. IP-MS data for PRR9-interacting proteomics are available via PtoteomeXchange with identifier PXD035252. Source data are provided with this paper.

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

## Acknowledgements

We would like to thank Dr. Rongcheng Lin (Institute of Botany, CAS) for the seeds of *cry1* and *cry2* mutants. We would like to thank Ms. Jingquan Li and Dr. Zhuang Lu from the Key Laboratory of Plant Molecular Physiology and Plant Science Facility of the Institute of Botany, CAS, for their technical assistance with confocal microscopy assay and IP-MS analysis, respectively. We appreciate Dr. JC Jang (Ohio State University) for his critical reading on the manuscript. The work was supported by the Strategic Priority Research Program of the Chinese Academy of Sciences, Grant No. XDB27030206 and National Natural Science Foundation of China (No. 31770287) to L.W.

## Author contributions

Y.H. and L.W. designed the project, analyzed the data, and wrote the paper. Y.Y. carried out a bioluminescence assay. X.W. carried out BiFC assay and vector construction. Y.Q. contributed to vector construction and plant genetic transformation. C.S. contributed to genetic material screening. L.W. agrees to serve as the author responsible for contact and ensures communication.

## Competing interests

The authors declare no competing interests.
