## [Peer Review File · Nature Communications]

Aschoff's Rules on Circadian Rhythms Orchestrated by Blue Light Sensor CRY2 and Clock Component PRR9Reviewer #1 (Remarks to the Author):

This manuscript focuses on the direct interaction between a member of the cryptochrome family, CRY2, and an element of the circadian clock core in plant, PRR9, under blue light conditions. The authors show that this physical interaction blocks the accessibility of PRR9 protein to its co-repressor TPL/TPRs and, consequently, its phosphorylation by PPKs kinases. Moreover, the phosphorylation of PRR9 influences its DNA binding capacity and its activity of repression of other important elements of the clock, such as CCA1. In my opinion, the manuscript presents very important results that shed light on the molecular communication between cryptochromes and the Arabidopsis clock core, regulating the response of the clock to the light stimulus in the entrainment process. This is one of the first molecular demonstrations of the role of cryptochromes in modulating the oscillations of the circadian clock in Arabidopsis, by acting directly on the clock core components.

The paper is clear and well written, the experiments are exhaustive and technically appropriate. For this reason, I believe that, pending some minor revisions listed below, the article may be acceptable for publication in Nature Communications.

Minor points

- In my opinion, the title is a bit misleading because it should focus on the interaction between CRY2 and PRR9 which is able to regulate the circadian clock, and not on Aschoff's rule;
- line 89- 90: "Moreover,, but against this rule in cB (Fig. 1B)". In my opinion this speculation is inaccurate because, at 2,5 μ m (Fig.1B) there is a large error bar that makes it impossible to state that the curve is not in conformity with Aschoff's rule;
- Figs 2B and 2C: lack of statistical analysis to confirm what is written in the main text;
- line 140: check typo "HDA6/19);
- Fig. 3D: lack of statistical analysis;
- line 169: it is necessary to explain the meaning of "zeitgeber time" and to add a reference;
- line 172: check typo "(Fig.3C),.";
- line 206: check typo "even (if)";
- Figs 6B, 6C, 6D and 6E: lack of statistical analysis;
- line 276: it could also be interesting to investigate the effect of the lack of PPK4;
- line 294: "indicating the regulation of circadian clock by PPKs at least in part requires PRR9 in cB"; I don't agree with this sentence: there are no relevant period differences between ppk2 ppk3 and ppk2 ppk3 prr9-1 mutants in cB;
- Figs 7A and 7B: lack of statistical analysis;
- line 329: check typo "rcadian";
- line 340: rewrite the sentence;
- line 349 and 365: I don't like "vital";
- line 405: check typo "tribble";
- line 509: describe the cycle used in "LD conditions";
- line 537: check the sentence;
- line 729: I don't like "specially",
- lines 745-746: check the sentence.

Reviewer #2 (Remarks to the Author):

Circadian clocks are autonomous timing mechanisms, but still, they need to be synchronized to the environmental day/night cycles to be useful to the host organism. Light is the most prominent environmental signal mediating this synchronization, called entrainment. One aspect of light-mediated entrainment is that light intensity affects the free-running period. Specifically, in diurnal organisms like plants, the higher the fluence rate of continuous illumination the shorter the free-running period. This general observation is known as Aschoff's rule. In plants, the molecular background of this phenomenon is not known yet. The present work provides a well-supported model explaining a part of this mechanism in Arabidopsis thaliana.

The Authors demonstrated that the core clock component PRR9 is bound by the blue light receptor CRY2 at low fluences of blue light. PRR9 is a transcriptional repressor and affects the function of the circadian oscillator via repressing the transcription of two other core clock components, CCA1 and LHY. The results showed that the CRY2-PRR9 interaction inhibits the function of PRR9 by at least two mechanisms: (i) it prevents the binding of PRR9 to its transcriptional co-repressor TPL, and (ii) it prevents the interaction of PRR9 with protein kinases PKKs, resulting in lower phosphorylation levels and thus, less efficient binding of PRR9 to chromatin around the CCA1 and LHY loci. The fluence rate sensitivity comes from the light-dependent (in)stability of CRY2. As blue light intensity increases, degradation of CRY2 is accelerated, and PRR9 is released from inhibition resulting in more efficient repression of CCA1/LHY and eventually in shorter periods of the clock. The results reveal a novel mechanism by which light can affect the function of the circadian clock. More precisely, the regulatory process described here accounts for a specific section of entrainment, which is mediated by CRY2-absorbed blue light. The report is logically structured and easy to read, and the English of the main text is good. The conclusions are supported by several layers of pieces of evidence, and the model summarising the findings is credible. Although I am basically positive about this manuscript, some concerns have been raised during reading it that should be addressed by the Authors.

Main issues

1. The effect of light fluence rate on the period is the central issue of this paper. Based on FRCs, the Authors state that *prr9-1* mutants do not obey Aschoff's rule in blue light - this is based on visual inspection and comparison of blue FRCs in *prr9-1* vs WT. It may be useful to provide a quantitative description of the slope of FRCs so they could be compared not just by eye.
2. Still related to FRCs: according to the final conclusion, *cry2* mutants should not obey Aschoff's law at lower fluence rates, which is seen in Fig.7A, indeed. However, right in the next panel (Fig.7B) graphs show that periods are shortened in the *cry2* mutant over the full range of fluences tested. This should be explained because showing these two contrasting results questions the creditability and reproducibility of data.
3. I think that the results concerning the genetic interaction of PRR9 and PKKs have been misinterpreted (lines 283-296). In my opinion, the period of the *pkk2 pkk3 prr9-1* triple mutant is comparable with that of the *pkk2 pkk3* double and is different from that of the *prr9-1* single. Strictly said, PKKs are epistatic to PRR9, which means that regulation of the circadian clock by PRR9 apparently requires PKKs, and not the other way around. This issue should be carefully discussed in light of the obvious regulatory relationships (i.e. PKKs phosphorylate PRR9, which affects its repressive function and eventually the speed of the clock).
4. The data indicate that CRY2 interacts with PRR9 in darkness, but not in red light (Fig.1F and H). Does red light actively inhibit the CRY2-PRR9 interaction? Please comment on this.

Minor issues

1. Please review Figure legends and describe all unexplained items. Even the obvious ones such as the red arrows in Fig5E and FigS1C.
2. The English of Materials and Methods could be improved. The main text (Intro, Results, Discussion) is written much better and contrasts the quality of the text of M&M.
3. I would like to see more technical details of the bioluminescence assays. Were individual seedlings or groups of seedlings assayed, what was the exposure time, what was the time interval between exposures, etc? The fact is that the conclusions of the ms are based on tiny but significant period differences that have been determined surprisingly precisely with extremely low variations/errors. I think that many of the researchers working on plant circadian biology routinely use luminescent reporters and they will be interested in how it is possible to reach such a low error rate.

Reviewer #3 (Remarks to the Author):

He et al. reported a novel mechanism for controlling the period length of the circadian clock under blue light. It has been known that light intensities influence the period lengths of the circadian oscillation. For example, higher light intensities shorten the circadian periods in plants. This characteristic of the circadian rhythm is known as Aschoff's rules. However, how the light signal

regulates the period lengths remained unknown. This current manuscript reports a new mechanism that takes a part in this regulation. They started with the previously characterized phenotype of the circadian clock mutant, *prr9*, which showed more pronounced clock phenotypes under blue light. The authors found that the PRR9 protein physically interacted with the CRY2 blue-light photoreceptor in a blue-light-dependent manner. Then they demonstrated that this interaction inhibited the repressive transcriptional activity of PRR9 by inhibiting the phosphorylation of PRR9 protein. They identified positions of phosphorylated amino acid residues of PRR9 protein under blue light using mass-spectrometry analysis. They followed this observation with functional assays using amino acid substitutions of the phosphorylated residues. Through this mutation analysis, the authors demonstrated that PRR9 protein with nine Ala substitutions of phosphorylated sites abolished the function of PRR9 protein. In addition, they identified the PPK kinases, which phosphorylate some of these residues. Furthermore, they showed that these kinases were genetically involved in the same light-dependent period-length controlling mechanisms. In total, the authors have found an exciting mechanism through their extensive experiments. Overall, the experiments were nicely executed, and the logic supporting each experiment is reasonable. Below are my comments.

Major comments:

1. On lines 225-261, the authors described the changes in the phosphorylation patterns (gel migration patterns) of PRR9 proteins with different amino acid substitutions. Although there was an apparent change in the migration pattern of PRR9(9A) when it was co-incubated with PPK1 (Fig. 5E), the patterns of PRR9 variants looked different when they were co-expressed with PPK2, PPK3, and PPK4 (Figs. S5C, D, and E). Especially with PPK2 and PPK3, the ratios of fast migrating bands and slow migrating bands look similar. Instead, some of the PRR9 variants showed different accumulation levels. Based on the results (shown in Fig. 5F and S5C, D, and E), what they discussed only fit the result with PPK1 (and maybe with PPK4). The authors should discuss the difference in PRR9 variant levels among PPKs. In this assay, the results looked like PPK2 and PPK3 can phosphorylate PRR9 residues other than these nine Ala-substituted residues. It was not clear whether these experiments were done in tobacco transient assay. The authors should indicate how these experiments were done.

2. This comment is also related to the previous one. On lines 265-267, the authors mentioned that their results in Fig. 5G supported the notion that CRY2 inhibits PPK1-dependent PRR9 phosphorylation. However, the results look like the overall amount of PRR9 became low when more CRY2 was expressed. To assess the authors' statement, quantifying slow and fast migrating bands may be necessary to describe the results more accurately.

3. In Fig. 5, they showed that four PPKs bound and phosphorylated PRR9. In Fig. 6, the authors presented the genetic results of the *pkk2 pkk3* double mutants. Why did the authors choose to show only the *pkk2 pkk3* mutant phenotype? Based on Figs. 5 and S5 results, it seems that PPK1 is more important, and PPK2 and PPK3 might be able to phosphorylate other residues than the ones the authors analyzed. They said in their methods that they obtained all single mutants from the stock center, but they did not show any circadian phenotypes of single mutants. Please explain the reason for choosing to present only the *pkk2 pkk3* mutants.

Minor comments:

1. ChIP results should be shown as % of input (%IP) rather than relative binding values.
2. Some results lack statistical analyses and should be treated with proper stats. These are Figs. 1B, 2B, 2C, 2D, 3D, 6B, 6C, 6D, 6E, 7A, and 7B.
3. Please indicate the protein sizes for each band in all western blot images.

Response to reviewers

Reviewer #1 (Remarks to the Author):

This manuscript focuses on the direct interaction between a member of the cryptochrome family, CRY2, and an element of the circadian clock core in plant, PRR9, under blue light conditions. The authors show that this physical interaction blocks the accessibility of PRR9 protein to its co-repressor TPL/TPRs and, consequently, its phosphorylation by PKKs kinases. Moreover, the phosphorylation of PRR9 influences its DNA binding capacity and its activity of repression of other important elements of the clock, such as CCA1. In my opinion, the manuscript presents very important results that shed light on the molecular communication between cryptochromes and the Arabidopsis clock core, regulating the response of the clock to the light stimulus in the entrainment process. This is one of the first molecular demonstrations of the role of cryptochromes in modulating the oscillations of the circadian clock in Arabidopsis, by acting directly on the clock core components.

The paper is clear and well written, the experiments are exhaustive and technically appropriate. For this reason, I believe that, pending some minor revisions listed below, the article may be acceptable for publication in Nature Communications.

Response: We appreciate your positive comments. We have addressed your concerns in detail as below.

Minor points

- In my opinion, the title is a bit misleading because it should focus on the interaction between CRY2 and PRR9 which is able to regulate the circadian clock, and not on Aschoff's rule;

Response: Thanks for your suggestion. In the revised manuscript, we have calculated the slope of FRC. According to the updated evidence and the labile nature of CRY2 protein with the increasing blue light intensity, we believe that our findings could fit the title. The interaction of CRY2 and PRR9 is a fact, while the biological significance of their interaction, together with their protein character, jointly provides a potential mechanism of Aschoff's rule. We appreciate your understanding on this issue very much.

-line 89- 90: "Moreover,....., but against this rule in cB (Fig. 1B)". In my opinion this speculation is inaccurate because, at 2,5 μ m (Fig.1B) there is a large error bar that makes it impossible to state that the curve is not in conformity with Aschoff's rule;

Response: We have repeated this experiment with more seedlings. We obtained similar results as shown in the revised Fig. 1b with smaller error bar. Based on the previous work by Farr, E.M. in 2005¹ and our current results, we thus state that curve of PRR9 in low blue light is not in conformity with Aschoff's rule.

-Figs 2B and 2C: lack of statistical analysis to confirm what is written in the main text;

Response: We have reanalyzed the data statistically and marked in Fig. 2b and 2c.

- line 140: check typo "HDA6/19);

Response: Fixed it, thanks.

- Fig. 3D: lack of statistical analysis;

Response: We have reanalyzed the data statistically and marked in Fig.3d.

-line 169: it is necessary to explain the meaning of "zeitgeber time" and to add a reference;

Response: We added a related reference in line 34-35 and give an explanation in Material and Method in line 567-569 of the revised manuscript.

-line 172: check typo "(Fig.3C),,,";

Response: Fixed it, thanks.

-line 206: check typo "even (if)";

Response: Fixed. Thanks.

-Figs 6B, 6C, 6D and 6E: lack of statistical analysis;

Response: We have reanalyzed the data in Figs 6b, 6c, 6d and 6e statistically and marked in the corresponding figures.

-line 276: it could also be interesting to investigate the effect of the lack of PPK4;

Response: We agree with you. Although the four PPKs members redundantly participate in many biological process, *ppk4* mutant is the only one which show long hypocotyl phenotype², suggesting its specific roles in hypocotyl elongation regulation. More genetic analysis will be done in the future.

-line 294: "indicating the regulation of circadian clock by PPKs at least in part requires PRR9 in cB"; I don't agree with this sentence: there are no relevant period differences between *ppk2 ppk3* and *ppk2 ppk3 prr9-1* mutants in cB;

Response: While considering previously reported that PPKs interacts with other circadian components, such as ELF3 and CCA1^{2,3}, we speculated that the circadian phenotype of PPKs at least in part requires PRR9. We added an explanation in line 299-303 in the revised manuscript.

-Figs 7A and 7B: lack of statistical analysis;

Response: We have reanalyzed the data statistically and marked in revised Fig. 7a and 7b.

-line 329: check typo "rcadian";

Response: Fixed, thanks.

-line 340: rewrite the sentence;

Response: We have rephrased it as "Nonetheless, our genetically evidence unambiguously supports that PRR9 and CRY2, but not CRY1, act in the same pathway to modulate circadian period" , as shown line 349-350 of the revised manuscript.

-line 349 and 365: I don't like "vital";

Response: We have changed it into "important" or "necessary" respectively.

-line 405: check typo "tribble";

Response: We have fixed this typo as "triple".

-line 509: describe the cycle used in "LD conditions";

Response: "LD condition (12 h light/12 h dark)" was added in line 500 of the revised manuscript..

-line 537: check the sentence;

Response: We have rephrased it as following " Finally, the protein-beads complex were resuspended and digested by elution buffer 1 (50 mM Tris/Cl pH 7.5; 2 M urea; 5 µg/ml sequencing grade modified trypsin; 1 mM DTT) at 30°C for 30 min. After centrifuging at 4°C, 3000 rpm for 2 min, the supernatant were transferred to new vial. The remaining beads were further resuspended by elution buffer 2 (50 mM Tris/Cl pH 7.5; 2 M urea; 5 mM iodoacetamide) for twice. The total supernatant from three times of elution were kept at 30°C overnight for digestion." in line 526-531 of the revised manuscript.

-line 729: I don't like "specially",

Response: “specially” is deleted in the revised manuscript.

-lines 745-746: check the sentence.

Response: We have changed it as “All Co-IP assays were repeated at least three times” in the revised manuscript.

Reviewer #2 (Remarks to the Author):

Circadian clocks are autonomous timing mechanisms, but still, they need to be synchronized to the environmental day/night cycles to be useful to the host organism. Light is the most prominent environmental signal mediating this synchronization, called entrainment. One aspect of light-mediated entrainment is that light intensity affects the free-running period. Specifically, in diurnal organisms like plants, the higher the fluence rate of continuous illumination the shorter the free-running period. This general observation is known as Aschoff's rule. In plants, the molecular background of this phenomenon is not known yet. The present work provides a well-supported model explaining a part of this mechanism in *Arabidopsis thaliana*.

The Authors demonstrated that the core clock component PRR9 is bound by the blue light receptor CRY2 at low fluences of blue light. PRR9 is a transcriptional repressor and affects the function of the circadian oscillator via repressing the transcription of two other core clock components, CCA1 and LHY. The results showed that the CRY2-PRR9 interaction inhibits the function of PRR9 by at least two mechanisms: (i) it prevents the binding of PRR9 to its transcriptional co-repressor TPL, and ⁴ it prevents the interaction of PRR9 with protein kinases PKKs, resulting in lower phosphorylation levels and thus, less efficient binding of PRR9 to chromatin around the CCA1 and LHY loci. The fluence rate sensitivity comes from the light-dependent (in)stability of CRY2. As blue light intensity increases, degradation of CRY2 is accelerated, and PRR9 is released from inhibition resulting in more efficient repression of CCA1/LHY and eventually in shorter periods of the clock.

The results reveal a novel mechanism by which light can affect the function of the circadian clock. More precisely, the regulatory process described here accounts for a specific section of entrainment, which is mediated by CRY2-absorbed blue light. The report is logically structured and easy to read, and the English of the main text is good. The conclusions are supported by several layers of pieces of evidence, and the model summarising the findings is credible. Although I am basically positive about this manuscript, some concerns have been raised during reading it that should be addressed by the Authors.

Response: We appreciate your overall positive comments and addressed your concerns one to one as below.

Main issues

1. The effect of light fluence rate on the period is the central issue of this paper. Based on FRCs, the Authors state that *prr9-1* mutants do not obey Aschoff's rule in blue light - this is based on visual inspection and comparison of blue FRCs in *prr9-1* vs WT. It may be useful to provide a quantitative description of the slope of FRCs so they could be compared not just by eye.

Response: Thanks for your helpful suggestion. We calculated the slope of FRC and added the quantitative description of Fig. 1b as following “Interestingly, when light intensity increased from 1 to 10 $\mu\text{mol m}^{-2} \text{s}^{-1}$, the slope of fluence response curve (FRC) for Col-0 ($k = -0.3$), *PRR9ox-1* ($k = -0.24$) and *prr9-1* mutant ($k = -0.07$) is in conformity with Aschoff's rule in cR, but *prr9-1* mutant is

against this rule in cB ($k = 0.22$) while the slopes of FRC for Col-0 ($k = -0.19$) and *PRR9ox-1* ($k = -0.3$) in cB is similar to in cR", from line 90-94 in the revised manuscript.

2. Still related to FRCs: according to the final conclusion, *cry2* mutants should not obey Aschoff's law at lower fluence rates, which is seen in Fig.7A, indeed. However, right in the next panel (Fig.7B) graphs show that periods are shortened in the *cry2* mutant over the full range of fluences tested. This should be explained because showing these two contrasting results questions the creditability and reproducibility of data.

Response: Thanks for your suggestion. In low blue light, *cry2* mutant exhibits long hypocotyl phenotype compared to Col-0, so it is hard to track their bioluminescent signals, which result in some seedlings with aberrant value of amplitude error. We have reanalyzed and updated Fig. 7a and b.

3. I think that the results concerning the genetic interaction of PRR9 and PKKs have been misinterpreted (lines 283-296). In my opinion, the period of the *pkk2 pkk3 prr9-1* triple mutant is comparable with that of the *pkk2 pkk3* double and is different from that of the *prr9-1* single. Strictly said, PKKs are epistatic to PRR9, which means that regulation of the circadian clock by PRR9 apparently requires PKKs, and not the other way around. This issue should be carefully discussed in light of the obvious regulatory relationships (i.e. PKKs phosphorylate PRR9, which affects its repressive function and eventually the speed of the clock).

Response: Sorry for this misleading. We agree that PRR9 and PPK2/PPK3 act in the same pathway to regulates circadian period. While considering previously reported that PPKs interacts with other circadian components, such as ELF3 and CCA1^{2,3}, we proposed that the circadian phenotype of PPKs at least in part requires PRR9. We added an explanation in line 299-302 in revised manuscript.

4. The data indicate that CRY2 interacts with PRR9 in darkness, but not in red light (Fig.1F and H). Does red light actively inhibit the CRY2-PRR9 interaction? Please comment on this.

Response: As reported CRY2 also interact with phytochrome B, in this case, we wonder PHYB may play a role in inhibiting the interaction between CRY2 and PRR9, which awaits to be tested in the future.

Minor issues

1. Please review Figure legends and describe all unexplained items. Even the obvious ones such as the red arrows in Fig5E and FigS1C.

Response: Thanks for your suggestion, we examined all of the figure legends and added the following information:

1. "b.i. stands for band intensity" in Fig. 2a and "Schematic diagram showing domain structure of PRR9 and position of phosphosites identified by IP-MS." in Fig.2g.

2. "Red open brace and green arrow indicated phosphorylated and unphosphorylated form of PRR9, respectively" in Fig. 3a and "Red and green arrow indicated phosphorylated and unphosphorylated form of PRR9, respectively" in Fig. 3c.

3. "Red arrow indicated phosphorylated PRR9" in Fig.5e and "Red triangle indicates increased CRY2 concentration." in Fig. 5g.

4. "The asterisk stands for nonspecific band and arrow indicates endogenous PRR9." in Fig. 6a.

5. "Red arrow indicates endogenous CRY2 recognized by CRY2 antibody." in Supplementary Fig. 1c.

2. The English of Materials and Methods could be improved. The main text (Intro, Results, Discussion) is written much better and contrasts the quality of the text of M&M.

Response: We have checked the section of Materials and Methods, and improved the language.

3. I would like to see more technical details of the bioluminescence assays. Were individual seedlings or groups of seedlings assayed, what was the exposure time, what was the time interval between exposures, etc? The fact is that the conclusions of the ms are based on tiny but significant period differences that have been determined surprisingly precisely with extremely low variations/errors. I think that many of the researchers working on plant circadian biology routinely use luminescent reporters and they will be interested in how it is possible to reach such a low error rate.

Response: Bioluminescence signals of each seedlings were individually captured and at least 15 seedlings were calculated. The parameter was set as 2-h interval for exposure and 30 mins dark for signal acquisition. According to your suggestions, we described the method of bioluminescence assays step by step and supplemented the details from Line 432-445 in the revised manuscript.

Reviewer #3 (Remarks to the Author):

He et al. reported a novel mechanism for controlling the period length of the circadian clock under blue light. It has been known that light intensities influence the period lengths of the circadian oscillation. For example, higher light intensities shorten the circadian periods in plants. This characteristic of the circadian rhythm is known as Aschoff's rules. However, how the light signal regulates the period lengths remained unknown. This current manuscript reports a new mechanism that takes a part in this regulation. They started with the previously characterized phenotype of the circadian clock mutant, *prr9*, which showed more pronounced clock phenotypes under blue light. The authors found that the PRR9 protein physically interacted with the CRY2 blue-light photoreceptor in a blue-light-dependent manner. Then they demonstrated that this interaction inhibited the repressive transcriptional activity of PRR9 by inhibiting the phosphorylation of PRR9 protein. They identified positions of phosphorylated amino acid residues of PRR9 protein under blue light using mass-spectrometry analysis. They followed this observation with functional assays using amino acid substitutions of the phosphorylated residues. Through this mutation analysis, the authors demonstrated that PRR9 protein with nine Ala substitutions of phosphorylated sites abolished the function of PRR9 protein. In addition, they identified the PPK kinases, which phosphorylate some of these residues. Furthermore, they showed that these kinases were genetically involved in the same light-dependent period-length controlling mechanisms. In total, the authors have found an exciting mechanism through their extensive experiments. Overall, the experiments were nicely executed, and the logic supporting each experiment is reasonable. Below are my comments.

Response: Thanks for your positive comments and we addressed your concerns one to one as below.

Major comments:

1. On lines 225-261, the authors described the changes in the phosphorylation patterns (gel migration patterns) of PRR9 proteins with different amino acid substitutions. Although there was

an apparent change in the migration pattern of PRR9(9A) when it was co-incubated with PPK1 (Fig. 5E), the patterns of PRR9 variants looked different when they were co-expressed with PPK2, PPK3, and PPK4 (Figs. S5C, D, and E). Especially with PPK2 and PPK3, the ratios of fast migrating bands and slow migrating bands look similar. Instead, some of the PRR9 variants showed different accumulation levels. Based on the results (shown in Fig. 5F and S5C, D, and E), what they discussed only fit the result with PPK1 (and maybe with PPK4). The authors should discuss the difference in PRR9 variant levels among PPKs. In this assay, the results looked like PPK2 and PPK3 can phosphorylate PRR9 residues other than these nine Ala-substituted residues. It was not clear whether these experiments were done in tobacco transient assay. The authors should indicate how these experiments were done.

Response: Thanks for your suggestion. The co-expression assay between PRR9 and PPKs were conducted in tobacco transient assay, we added the information in the corresponding figure legends.

In consistent with your opinion, we added the results description “The results showed that PPK1 and PPK4 display similar roles in catalyzing PRR9 phosphorylation, for the band shift promoted by PPK1 and PPK4 almost disappeared when they were coexpressed with PRR9^{9A} compared to PRR9 and other phosphosite mutations (Fig. 5f and Supplementary Fig. 5e). While the phosphorylation pattern is slightly different from PPK2 and PPK3, as we observed PPK2 and PPK3 promoted specific band shift migrates slightly faster when coexpressed with PRR9^{9A} and PRR9^{S267A/S269A} (Supplementary Fig. 5c, d), suggesting PPK2 and PPK3 may recognize other unidentified residues besides S267 S269.” in line 257-265 of revised manuscript.

Meanwhile, we added “We observed differential accumulation of phosphorylated PRR9 between PRR9 variants when they were co-expressed with PPKs, suggesting some phosphosites are associated with phosphorylated PRR9 protein stability, while the underlying mechanisms waits for further exploring” in line 361-365 of revised manuscript.

2. This comment is also related to the previous one. On lines 265-267, the authors mentioned that their results in Fig. 5G supported the notion that CRY2 inhibits PPK1-dependent PRR9 phosphorylation. However, the results look like the overall amount of PRR9 became low when more CRY2 was expressed. To assess the authors’ statement, quantifying slow and fast migrating bands may be necessary to describe the results more accurately.

Response: In consistent with your opinion, the overall amount of PRR9 indeed became low when coexpressed with CRY2. As suggested, we measured the band intensity of phosphorylated/unphosphorylated PRR9 and added the ratio in revised Fig. 5g. The results support our notion that CRY2 inhibits PPK1-dependent PRR9 phosphorylation.

3. In Fig. 5, they showed that four PPKs bound and phosphorylated PRR9. In Fig. 6, the authors presented the genetic results of the *ppk2 ppk3* double mutants. Why did the authors choose to show only the *ppk2 ppk3* mutant phenotype? Based on Figs. 5 and S5 results, it seems that PPK1 is more important, and PPK2 and PPK3 might be able to phosphorylate other residues than the ones the authors analyzed. They said in their methods that they obtained all single mutants from the stock center, but they did not show any circadian phenotypes of single mutants. Please explain the reason for choosing to present only the *ppk2 ppk3* mutants.

Response: According to our biochemical results, PPK1 is indeed more important than PPK2/PPK3 to phosphorylate PRR9. However, the circadian phenotypes of PPK single mutants have been reported², that *ppk1* has no circadian phenotype, while *ppk2*, *ppk3* and *ppk4* single mutant all

exhibits long free running period. Besides, the circadian period of *ppk1 ppk2*, *ppk1 ppk4* and *ppk1 ppk2 ppk4* double or triple mutants are similar to wild type ², meaning *ppk1* can restore the circadian phenotype of other *PPKs* to the wild type. Hence, we choose the evolutionary closed *PPK2* and *PPK3* to analyze the circadian phenotype. We have added the explanation in line 277-282 of revised manuscript.

Minor comments:

1. ChIP results should be shown as % of input (%IP) rather than relative binding values.

Response: We have revised the ChIP results of Fig.2e and Fig.4e as suggested.

2. Some results lack statistical analyses and should be treated with proper stats. These are Figs. 1B, 2B, 2C, 2D, 3D, 6B, 6C, 6D, 6E, 7A, and 7B.

Response: We have analyzed these data statistically and marked in the corresponding figures, thanks.

3. Please indicate the protein sizes for each band in all western blot images.

Response: Thanks for your suggestion. We have added these information.

- 1 Farre, E. M., Harmer, S. L., Harmon, F. G., Yanovsky, M. J. & Kay, S. A. Overlapping and distinct roles of PRR7 and PRR9 in the Arabidopsis circadian clock. *Curr Biol* **15**, 47-54 (2005).
- 2 Huang, H. *et al.* Identification of Evening Complex Associated Proteins in Arabidopsis by Affinity Purification and Mass Spectrometry. *Mol Cell Proteomics* **15**, 201-217 (2016).
- 3 Zheng, H. *et al.* MLK1 and MLK2 coordinate RGA and CCA1 activity to regulate hypocotyl elongation in Arabidopsis thaliana. *Plant Cell*, **30**, 67-82 (2018).
- 4 Okabe, Y. *et al.* Tomato TILLING technology: development of a reverse genetics tool for the efficient isolation of mutants from Micro-Tom mutant libraries. *Plant Cell Physiol* **52**, 1994-2005 (2011).

Reviewer #1 (Remarks to the Author):

The authors significantly improved the manuscript, which is now suitable for publication in Nature Communications.

Paolo Facella

Reviewer #2 (Remarks to the Author):

This is the revised version of a manuscript I reviewed earlier. In the current version, the Authors have addressed all of my concerns adequately and modified the manuscript according to my suggestions. I am satisfied with the Authors' reply to my questions and accept the explanations they provided. I am convinced that no further changes to the ms are required.

Reviewer #3 (Remarks to the Author):

In this revised manuscript, the authors responded to most of my previous comments successfully. However, I found that the stats they applied for some of their results seemed to be inappropriate. This point needs to be fixed.

For example, in response to Reviewer 1 comment, the authors repeated the experiments shown in Fig. 1b. They said they ran a one-way ANOVA followed by Fisher's LSD test for the results. The same letters in the figure denote no differences among them. Based on the information in the supplemental table, the authors seemed to compare the period length among different genotypes (Col-0, prr9-1, and PRR9ox) at each light condition. This is not what the authors intended to discuss. If that is what they would like to discuss, they should separate the graph into each light intensity. Instead, they discussed by comparing the slopes between different period lengths (2.5 and 10 $\mu\text{mol}/\text{m}^2/\text{sec}$) in each line under either R or B light (text lines 89-93 – there is a typo in line 90, it should be "from 2.5 to 10.") The authors must choose the proper stats based on what they would like to discuss. For the same reason, the proper stats need to be applied for Fig. 7a and b. Also for gene expression results shown in 2b, 2c, and 3d. There are more than two lines/conditions at each time point. The authors used the Student's t-test to compute p-values. They need to specify which pair they are comparing. I strongly suggest the authors consult a biostatistician to apply proper stats for all of their results.

Because Fig. 1 b is the only figure that shows the prr9 period length response under different blue light intensities (2.5-10 $\mu\text{mol}/\text{m}^2/\text{sec}$) might not fit the Aschoff's rule, and based only on one condition and the improper stat analysis, I am not convinced the statement. For example, the slope of the prr9 period length differences between 2.5 and 10 $\mu\text{mol}/\text{m}^2/\text{sec}$ in R also looks similar to the B one. Maybe the authors should remove the words "Aschoff's rule" from the title.

For new ChIP-result graphs (2e and 4e), they used box and whisker plots. It is not suitable to use the box and whisker plots for the $n=3$ results. The number ($n=3$) is too few to show higher and lower quartiles, maximums, and minimums. Please use regular bar graphs superimposed with individual data values.

REVIEWER COMMENTS

Reviewer #1 (Remarks to the Author):

The authors significantly improved the manuscript, which is now suitable for publication in Nature Communications.

Response: Thanks again for your helpful comments.

Reviewer #2 (Remarks to the Author):

This is the revised version of a manuscript I reviewed earlier.

In the current version, the Authors have addressed all of my concerns adequately and modified the manuscript according to my suggestions. I am satisfied with the Authors' reply to my questions and accept the explanations they provided. I am convinced that no further changes to the ms are required.

Response: We appreciate your constructive comments to improve our manuscript.

Reviewer #3 (Remarks to the Author):

In this revised manuscript, the authors responded to most of my previous comments successfully. However, I found that the stats they applied for some of their results seemed to be inappropriate. This point needs to be fixed.

Response: Sorry for the undefined information about statistical analysis. We have addressed your concerns one by one as below.

For example, in response to Reviewer 1 comment, the authors repeated the experiments shown in Fig. 1b. They said they ran a one-way ANOVA followed by Fisher's LSD test for the results. The same letters in the figure denote no differences among them. Based on the information in the supplemental table, the authors seemed to compare the period length among different genotypes (Col-0, *prr9-1*, and *PRR9ox*) at each light condition. This is not what the authors intended to discuss. If that is what they would like to discuss, they should separate the graph into each light intensity. Instead, they discussed by comparing the slopes between different period lengths (2.5 and 10 $\mu\text{mol}/\text{m}^2/\text{sec}$) in each line under either R or B light (text lines 89-93-there is a typo in line 90, it should be "from 2.5 to 10.") The authors must choose the proper stats based on what they would like to discuss.

Response: Thanks for correcting us. Followed your suggestion, we have re-performed the statistical analysis by comparing the circadian period length of same genotypes (Col-0, *prr9-1*, and *PRR9ox*) among different light intensity.

For the same reason, the proper stats need to be applied for Fig. 7a and b.

Response: In fig.7a and 7b, we tended to compare the circadian period length of different genotypes at each light condition to show the genetic relationship between PRR9 and CRY2. We have added the corresponding explanations into figure legends of the revised manuscript to avoid misunderstanding.

Also for gene expression results shown in 2b, 2c, and 3d. There are more than two lines/conditions at each time point. The authors used the Student's t-test to compute p-values. They need to specify which pair they are comparing. I strongly suggest the authors consult a biostatistician to apply proper stats for all of their results.

Response: Thanks for your helpful suggestion. We have changed the line graph of fig. 2b, 2c and 3d into bar graphs. We believe the statistical analysis between the compared data groups now were clearly displayed.

Because Fig. 1 b is the only figure that shows the *prp9* period length response under different blue light intensities (2.5-10 $\mu\text{mol}/\text{m}^2/\text{sec}$) might not fit the Aschoff's rule, and based only on one condition and the improper stat analysis, I am not convinced the statement. For example, the slope of the *prp9* period length differences between 2.5 and 10 $\mu\text{mol}/\text{m}^2/\text{sec}$ in R also looks similar to the B one. Maybe the authors should remove the words "Aschoff's rule" from the title.

Response: Thanks for your suggestion. Besides of our data, the circadian periods of *prp9* and *prp7* in a range of blue and red light intensity have been tested with taking *CCR2:LUC* as a circadian reporter (Farre et al., Current Biology, 2005). Their evidence also demonstrated that the fluence response curve for *prp9-1* resembles that of the *cry1 cry2* mutant. Similarly, we also showed the FRC of *prp9-1* is similar to that of *cry2* mutant in Fig. 7a with *CCA1:LUC* as a circadian reporter. Moreover, the slope of *prp9-1* in cB is 0.66 while in cR is -0.36, as shown in the revised Fig. 1b. In addition, we think "Aschoff's rule" in the title may arrive to more broad readers. We appreciate your understandings.

For new ChIP-result graphs (2e and 4e), they used box and whisker plots. It is not suitable to use the box and whisker plots for the n=3 results. The number (n=3) is too few to show higher and lower quartiles, maximums, and minimums. Please use regular bar graphs superimposed with individual data values.

Response: Thanks for your suggestion. We have updated Fig 2e and 4e as suggested.

Reviewer #3 (Remarks to the Author):

The authors responded to this reviewer's comments satisfactorily. I don't have any further comments on this manuscript.